# Study on mechanical properties of corroded concrete columns strengthened with SMA wires

Qiang Pei[ID][1]*, Bangwen Cai[1], Zhicheng Xue[2], Yu Ding[1], Di Cui[3], Yun Guo[1]

1 College of Architectural Engineering, Dalian University, Dalian, China, 2 Architecture and Civil Engineering Institute, Guangdong University of Petrochemical Technology, Maoming, China, 3 School of Civil Engineering, Dalian Jiaotong University, Dalian, China

* pqiem@163.com

## Abstract

Ocean crossing bridges suffer from seawater corrosion all year round and their mechanical properties will be substantially diminished. In order to enhance the mechanical properties of reinforced concrete columns corroded by seawater, SMA wire is used to restrain the reinforced concrete columns corroded by seawater to study their mechanical properties. 14 specimens were produced through the test, and the natural seawater corrosion was simulated by preparing a certain concentration of synthetic seawater. The mechanical properties of SMA strengthened specimens and unreinforced specimens are compared and analyzed, including failure mode, hysteresis curve, bearing capacity, ductility, stiffness and energy dissipation; the effects of different synthetic seawater corrosion concentrations on the mechanical properties of reinforced concrete columns are discussed. The results show that the bearing capacity and stiffness of reinforced concrete columns subjected to synthetic seawater corrosion are substantially diminished than those of uncorroded specimens, and the bearing capacity of specimens decreases more with the increase of synthetic seawater corrosion concentration; synthetic seawater corrosion has obscure effect on the ductility and energy dissipation performance of the specimens. The mechanical properties of the corroded specimens strengthened with SMA wire have been substantially enhanced, particularly the energy dissipation performance and bearing capacity have been notably enhanced, and the ductility and stiffness have also been somewhat enhanced. At the same time, based on the test, the finite element model is created according to the test specimen, while the accuracy of the model is verified, and the effects of the spacing and diameter of SMA wire, the strength of concrete and the thickness of protective layer on the mechanical properties of the specimen are analyzed.

## Introduction

Reinforced concrete columns as part of offshore bridges and other infrastructure have their own advantages, such as high bearing capacity, good durability and compactness. Nonetheless,

**Data Availability Statement:** All relevant data are within the article.

**Funding:** This research was supported by the National Natural Science Foundation of China (Grant Nos. 51878108, 52032011), Department of Science & Technology Guidance Plan Foundation

of Liaoning Province (Grant Nos. 2019JH8/
10100091), Scientific Research Project of Liaoning
Provincial Department of Education (Grant Nos.
LJKZ1177). The funders had no role in study
design, data collection and analysis, decision to
publish, or preparation of the manuscript.

**Competing interests:** The authors have declared
that no competing interests exist.

the coastal sea crossing bridges are invariably in a harsh service environment with seawater erosion, wave impact and freeze-thaw action for a long time. In addition, it also will be subjected to the vehicle's vertical reciprocating load and dead weight of the superstructure above the pier. The dynamic coupling of these actions will diminish the durability, bearing capacity and overall mechanical properties of reinforced concrete column [1–3], and subsequently will accelerate its damage and even threaten the safety of the structure [4, 5]. Recent research has disclosed that the flexural capacity and ductility of pier sections of coastal bridges also will be diminished when subjected to the coupling actions of seawater corrosion and reciprocating load for various decades [6]. On the one hand, when the concrete of a reinforced concrete column is corroded by seawater, chloride and sulfate in seawater will penetrate into the concrete and then the internal rebar will be rusted by strong corrosion ions and consequently the effective bearing area of reinforcement is tiny, eventually resulting in the reduction of overall bearing capacity of the column [7–9]. On the other hand, when the concrete is corroded by chloride and acid, its dynamic compressive strength and impact toughness will decrease significantly. Meanwhile, many pores will appear in the concrete just because of the seawater corrosion, as its energy dissipation capacity will seriously be weakened [10–12]. On the third hand, after reinforced concrete was corroded the bond mechanical properties between concrete and rebar will be diminished, which will decline the overall mechanical properties and strength of reinforced concrete structure [13–17]. Structural performance degradation caused by seawater corrosion is a common phenomenon, even the protected concrete-filled steel cylindrical structure is no exception. Some researchers have verified that under the combined action of long-term seawater corrosion and reciprocating load, the axial tensile bearing capacity, ductility and stiffness of concrete-filled steel cylindrical will decline somewhat [18–21]. And this phenomenon is more obvious in concrete-filled steel cylindrical short columns. Moreover, with the increase of corrosion time and load, the reduction degree will increase sharply [22]. At the same time, after the concrete-filled steel tube is corroded, its hysteresis curve changes from full to shuttle, that is to say, its energy dissipation performance is also weakened [23–25]. In general, seawater corrosion will cause many adverse effects and serious consequences to marine engineering structures, such as weakened mechanical properties, diminished service life and so on. Accordingly, we must pay attention to these adverse effects caused by corrosion and take some measures to prevent some disastrous consequences.

Consolidation is an important way to enhance the durability and other properties of reinforced concrete columns. The commonly used reinforcement measures include increasing section, pasting FRP cloth and prestressing externally. The method of bonding FRP cloth has the advantages of light weight, strong corrosion resistance and high strength; it is brittle unfortunately. External processing is a method to install the prestressed steel wire, steel strand and other metal materials on the structural surface, which can prevent the cracking of concrete without section increase, delay the penetration to concrete of chloride and sulfate in seawater and hinder the corrosion of rebar in the marine environment. The shape memory alloy (SMA) is a new intelligent material that takes into account sensing and driving functions. It is commonly divided into three categories according to alloy types, namely Ni-Ti-based, Cu-based, and Fe-based, respectively. Different families of shape memory alloys have different application ranges, advantages and disadvantages because of their different transformation temperatures and mechanical properties [26]. Cu-based shape memory alloys have superior thermal conductivity and electrical conductivity, and are frequently used in the field of mechanical manufacturing [27], but its application is hindered by unstable memory property and poor fatigue resistance. Fe-based shape memory alloys have been widely used in the field of structural welding due to its superior welding property and low cost [28]. Ni-Ti shape memory alloys have been in the leading position in the application of memory alloys. It has the

characteristics of super elasticity, strong self recovery ability, and excellent fatigue resistance. When the external force is unloaded, the inverse phase transformation drive can automatically restore the strain up to 8%–10% instantaneously [29]. At present, Ni-Ti shape memory alloy has been widely studied and applied in energy dissipation braces [30, 31], earthquake resistance [32–34], isolation bearing [35, 36], and various dampers [37–39]. Meanwhile, some research finished in the past has shown that the bearing capacity [40, 41], deformation capacity and axial mechanical properties [42] of reinforced concrete columns strengthened with SMA are significantly improved. In addition, to embed SMA tendon in reinforced concrete beams and columns can better enhance their deformation and bearing capacity [43–45]. The comparative study between thin-walled steel tube and SMA strengthened reinforced concrete columns indicates that SMA can better improve the energy dissipation performance of columns and enhance the axial compression performance of specimens [46, 47]. The above is a traditional application based on the properties of SMA. In recent years, some researchers have applied SMA to modern structures, and retuned the tuned mass damper by adjusting the temperature of SMA to reduce the wind-induced vibration and reduce the structural response [48, 49].

To sum up, the above research results predominantly focus on single corrosion environment and monotonic loading on concrete columns, and the data and conclusions are still limited. Nevertheless, while the actual coastal bridges are eroded by seawater, the piers will likewise be subject to the coupling effect of vertical reciprocating loads of different load vehicles and potential vertical seismic loads beyond the design basis for a long time, which further aggravates the risk of damage to the bridge structure during operation. Based on this, in order to ensure the operation safety of the bridge in the harsh corrosive environment such as seawater and under the action of reciprocating dynamic load, this paper studies the reinforcement and repair effect of the bridge pier column after seawater corrosion. The reinforced concrete column is corroded by prepared salt solution to simulate seawater corrosion, and the corrosion by salt solution is strengthened with SMA wire, so as to study the mechanical properties and reinforcement effect of reinforced concrete column under uniaxial reciprocating load, to provide reference for the reinforcement design of in-service coastal engineering after seawater erosion.

## Experimental program

### Test specimen

In order to study the effects of artificial seawater concentration and before and after reinforcement on the mechanical properties of reinforced concrete short columns under artificial seawater corrosion, fourteen reinforced concrete short columns were designed. In the name of the test piece, C is the acronym of the word "concrete", 1 represents the first group of test samples, W is the acronym of the word "without", S is the acronym of "SMA", and the last number 1 or 2 represents the order of a group of samples. For example, C1W-1 represents the first unreinforced concrete column in the first group, and C3S-1 represents the first SMA reinforced concrete column in the third group. See Table 1 for details.

All test specimens are circular reinforced concrete columns with a diameter of 150 mm and a height of 300 mm. The strength grade of concrete is C50 [50] and the thickness of protective layer is 10 mm. Four HRB400 with a diameter of 12 mm are used for longitudinal reinforcement and 6 mm for stirrup. The detailed structure of the test piece is shown in Fig 1.

### Material properties

**SMA materials.**   The Ni-Ti shape memory alloy wire used in this test was provided by Xi'an Zhongtai New Material Technology Co., Ltd. The chemical composition is as follows:

**Table 1. Specimens design.**

| Specimen number | Is it reinforced Yes or No | Reinforcement measures | Salt solution concentration |
|---|---|---|---|
| C1W-1 | No | - - - | 0% |
| C1W-2 | No | - - - | 0% |
| C2W-1 | No | - - - | 15% |
| C2W-2 | No | - - - | 15% |
| C3S-1 | Yes | SMA | 15% |
| C3S-2 | Yes | SMA | 15% |
| C4W-1 | No | - - - | 20% |
| C4W-2 | No | - - - | 20% |
| C5S-1 | Yes | SMA | 20% |
| C5S-2 | Yes | SMA | 20% |
| C6W-1 | No | - - - | 25% |
| C6W-2 | No | - - - | 25% |
| C7S-1 | No | SMA | 25% |
| C7S-2 | No | SMA | 25% |

Ni: 55.96%, Ti: 44.14%. The diameter and length of the Ni-Ti alloy wire are 1.2mm and 250mm, respectively, and the gauge distance is 150mm.

The cyclic tensile test of SMA wire was carried out according to the strain amplitude (1%, 2%,. . ., 8%). The loading rate is fixed at 30mm per minute, and the test room temperature was 20°C. The loading device of tensile test is shown in Fig 2. The properties of the Ni-Ti alloy after the tensile test are shown in Table 2. Since the influence of temperature is not considered in this paper, the transition temperature of Ni-Ti SMA wire is not tested, only the transition temperature provided by the manufacturer, as shown in Table 3.

**Reinforced steel and making salt solution.** The information of reinforcement used in this test is as follows: the longitudinal reinforcement is HRB400 with a diameter of 12mm, and the stirrup is HPB300 with a diameter of 6mm. See Table 4 for its mechanical properties.

To accelerate the corrosion process and shorten the test cycle. In this paper, 10 times, 13 times and 17 times of natural seawater concentration are used as artificial seawater for indoor corrosion test, corresponding to 15%, 20% and 25% of salt solution concentration, respectively. The concentrations of components in natural seawater and 15% salt solution are shown in Tables 5 and 6 respectively.

## Reinforced concrete column strengthened with SMA wire

When strengthening concrete columns, this test used a wire tensioning and anchoring device to strengthen SMA wires on concrete columns. The device comprises two clamps for fixing the

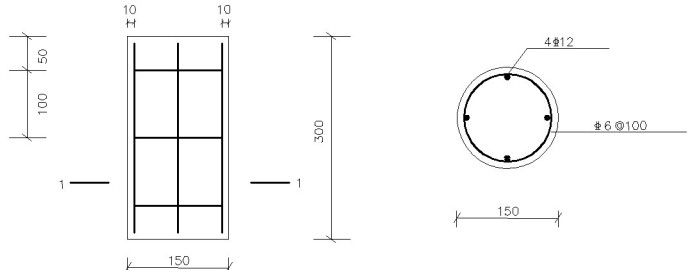

**Fig 1. Dimension drawing of test column.**

(a)

(b)

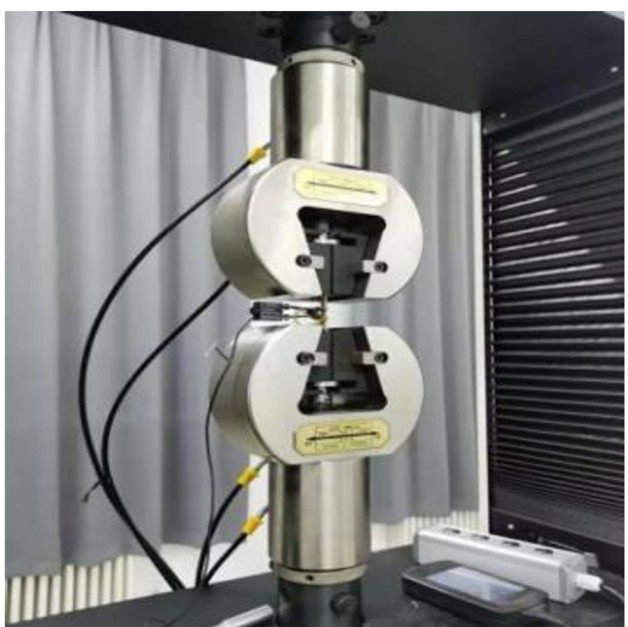
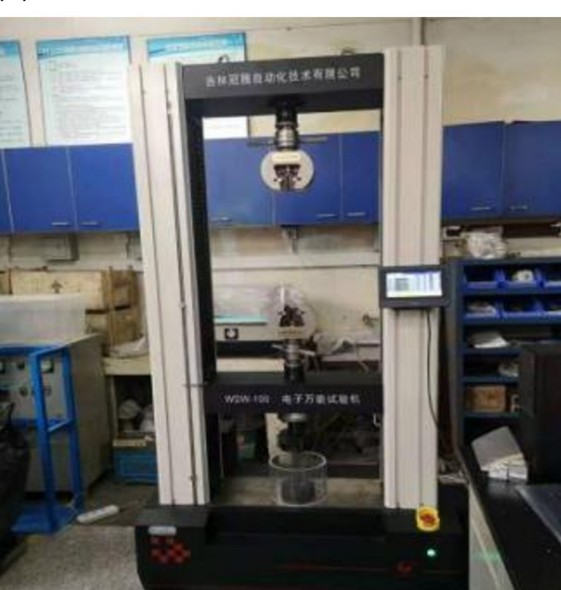

**Fig 2. Tensile test device.**

**Table 2. SMA mechanical properties.**

| Alloy material | Density (kg.m$^{-3}$) | Elastic modulus/GPa | Tensile strength/MPa | Yield strength/MPa | Recovery strain/% |
|---|---|---|---|---|---|
| Ni-Ti alloy | 7800 | 65.8 | 610 | 95 | 4.3 |

**Table 3. SMA wire transition temperature.**

| As[˚C] | Af[˚C] | Ap[˚C] | Ms[˚C] | Mf[˚C] | Mp[˚C] |
|---|---|---|---|---|---|
| -21 | -9 | -14 | -36 | -51 | -42 |

**Table 4. Mechanical properties of steel bar.**

| Reinforcement type | Diameter/mm | Yield strength $f_y$ /MPa | Tensile strength $f_u$ /MPa | Elastic modulus $E_s$/($10^5$MPa) | Yield strain $\varepsilon_y$ /$10^{-6}$ |
|---|---|---|---|---|---|
| Longitudinal steel bar | 12 | 415.19 | 604.93 | 2.07 | 2006 |
| | 6 | 364.85 | 546.88 | 2.07 | 1763 |

**Table 5. Content of various components in natural seawater.**

| Ingredients | NaCl | MgCl$_2$ | MgSO$_4$·7H$_2$O | CaSO$_4$·2H$_2$O | CaCO$_3$ |
|---|---|---|---|---|---|
| Concentration(g/L) | 21 | 2.51 | 1.54 | 2.43 | 0.1 |

**Table 6. Concentration of each component of 15% salt solution.**

| Ingredients | NaCl | MgCl$_2$ | MgSO$_4$·7H$_2$O | CaSO$_4$·2H$_2$O | CaCO$_3$ |
|---|---|---|---|---|---|
| Concentration(g/L) | 210 | 25.1 | 15.4 | 24.3 | 1 |

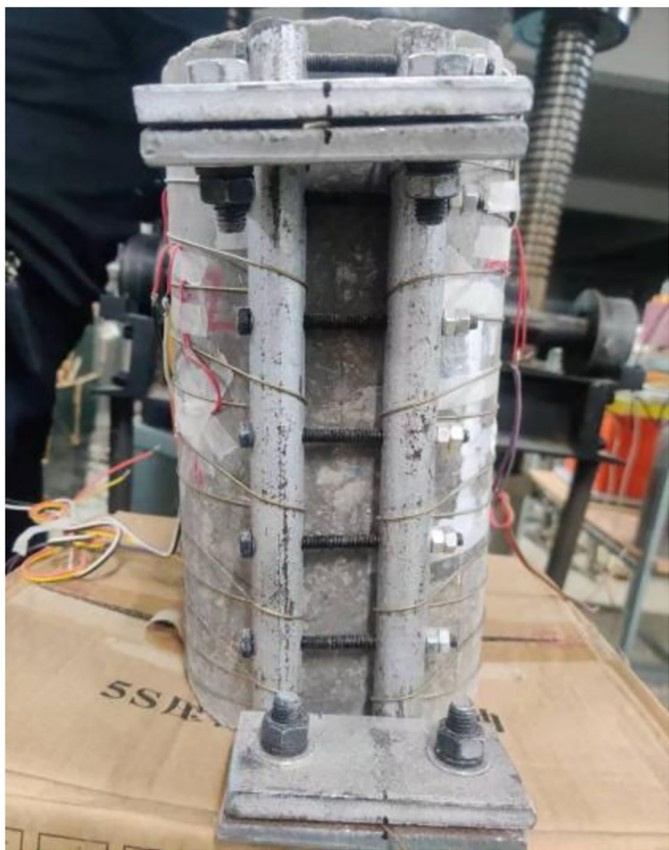

**Fig 3. SMA wire reinforced column.**

ends of SMA wires with a length of 10cm and a width of 5cm, two steel bar prestressed anchors with a diameter of 20mm, and seven prestressed high-strength bolts with a diameter of 6mm. The concrete column strengthened with SMA wire is shown in Fig 3.

## Test loading and measurement

The loading device is a 2000kN hydraulic pressure testing machine, which is composed of a press, an oil pump system and a control system. Fig 4 shows the axial compression test device and the corresponding loading diagram, respectively.

The quantities tested in the test mainly include: transverse strain of concrete column (upper 1 / 4 position and middle 1 / 2 position), vertical displacement and axial compression load. The strain and other data are collected by the static strain gauge, and the load and vertical displacement are obtained by the pressure sensor and the calibrated yhd-50 displacement sensor.

This test is a static load test under unidirectional cyclic loading. According tothe relevant provisions in the standard for test methods of concrete structures (GB50152-2012). The loading is completed by stages by means of displacement loading, which is divided into two parts: preloading and main loading: the function of preloading is to check whether the sensor, strain gauge, and loading equipment are normal; whether the reinforced concrete column specimen is in good contact with the force sensor and testing machine; whether the position alignment of reinforced concrete column specimen is accurate to avoid bias loading and uneven stress.

(a)                                                          (b)

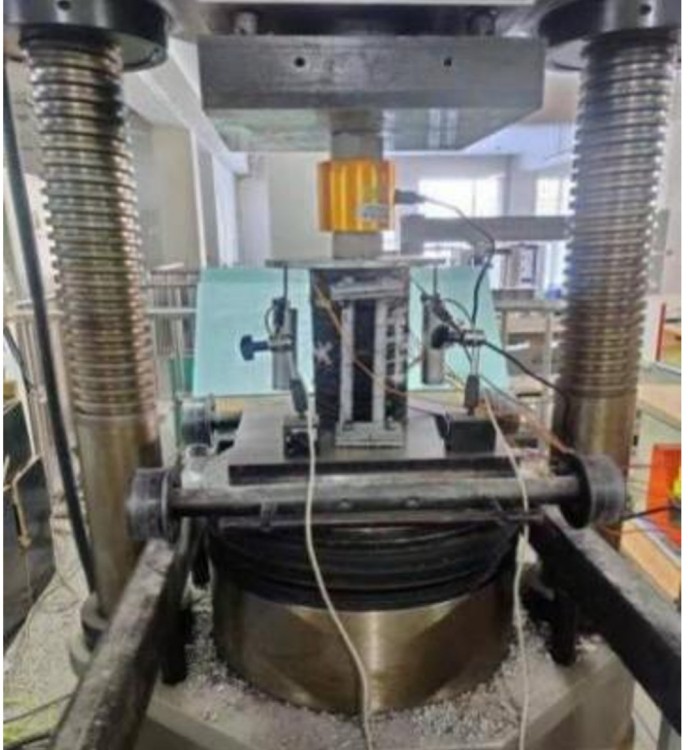
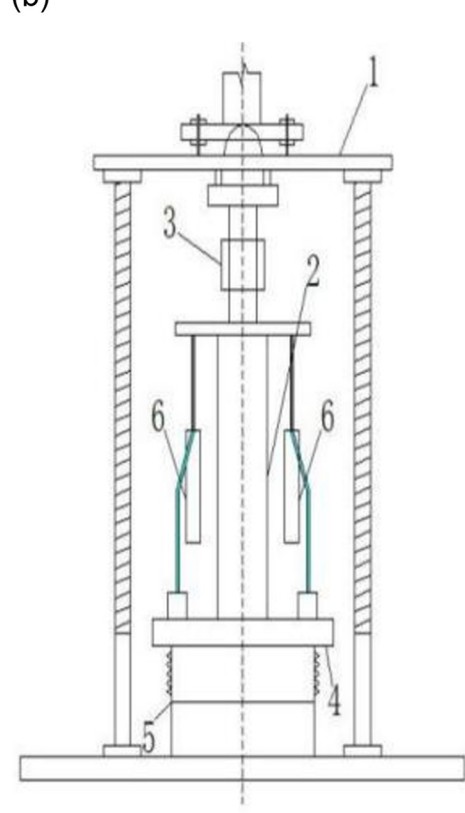

**Fig 4. Test device.** 1-Upper pressing plate of testing machine. 2-Testing column. 3-Load sensor. 4-Lower pressing plate of testing machine. 5-Testing machine cylinder. 6-Displacement sensor.

During main loading, the increment of displacement grading loading was 0.5mm, and each stage was cycled three times, until the specimen is destroyed, as shown in Fig 5.

## Analysis of test results

### Test phenomenon and failure mode

Fig 6 shows the failure mode of each test piece. The test phenomena of unreinforced specimens are basically similar, taking the C1W-1 specimen as an example. When the displacement is

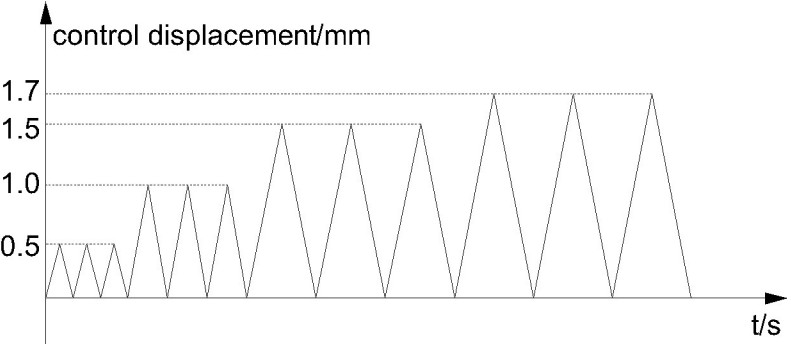

**Fig 5. Loading protocol.**

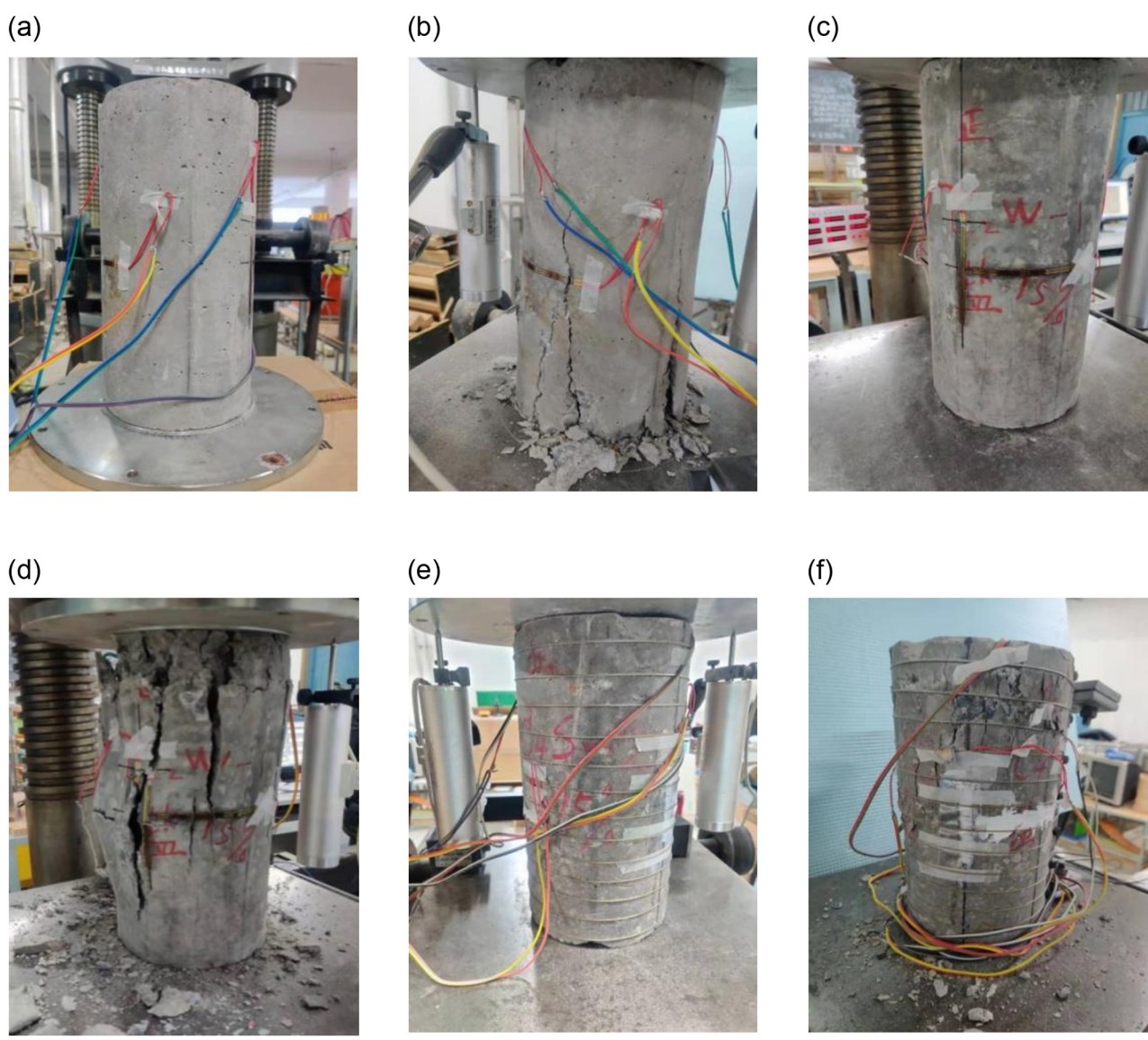

**Fig 6. Reinforced concrete specimens before and after the test.** (a) C1W-1 Before test. (b) C1W-1 After test. (c) C2W-1 Before test. (d) C2W-1 After test. (g) C3S-1 Before test. (h) C3S-1 After test.

loaded to 3mm, the load reaches 328.06kN, micro cracks appear at the bottom of the test piece, and a small amount of concrete fragments peel off on the surface of the column body. When the ultimate bearing capacity is reached, the bearing capacity decline rapidly, and the cracks evolve rapidly. Basically, a vertical through joint is formed in the middle and upper part of the test piece. There are a large number of peeling concrete fragments and obvious bulges in the middle and upper part of the test piece. The whole specimen shows multiple cracks, and the upper and lower parts form local swelling, which eventually fails to bear the force.

The test phenomena of reinforced specimens are basically similar, taking the C3S-1 specimen as an example. When the displacement is loaded to 5mm, the load reaches 222.14kN, and there are fine cracks on the top of the specimen, accompanied by the spalling of a small amount of concrete fragments. When the ultimate bearing capacity is reached, the bearing capacity decreases slowly, followed by the reverse growth, and another peak bearing capacity

appears, forming a stable bearing, and the original cracks are basically unchanged; when four displacement levels are continuously loaded, the load value slowly decreases to 271.4kN, and a large number of concrete fragments peel off on the surface of the specimen. At this time, the bearing capacity of the specimen decreases rapidly, the deformation is obvious, the SMA wire is broken and the specimen is damaged. Compared with the unreinforced specimen, the final failure of the SMA wire reinforced specimen did not form multiple through cracks and local swelling, and only a small number of local micro cracks, which is because the SMA wire reinforcement improves the bearing capacity and deformation capacity of the concrete column.

## Load displacement curve

The load displacement relationship curves of specimens without reinforcement and with reinforcement are shown in Fig 7. As can be seen from Fig 7a and 7b, the ultimate bearing capacity of the uncorroded and unreinforced specimen (C1W-1) reaches 388kN, while the ultimate bearing capacity of the corroded and unreinforced specimen (C2W-1) is only 285kN, indicating that the bearing capacity of the specimen corroded by seawater is reduced, but the ductility

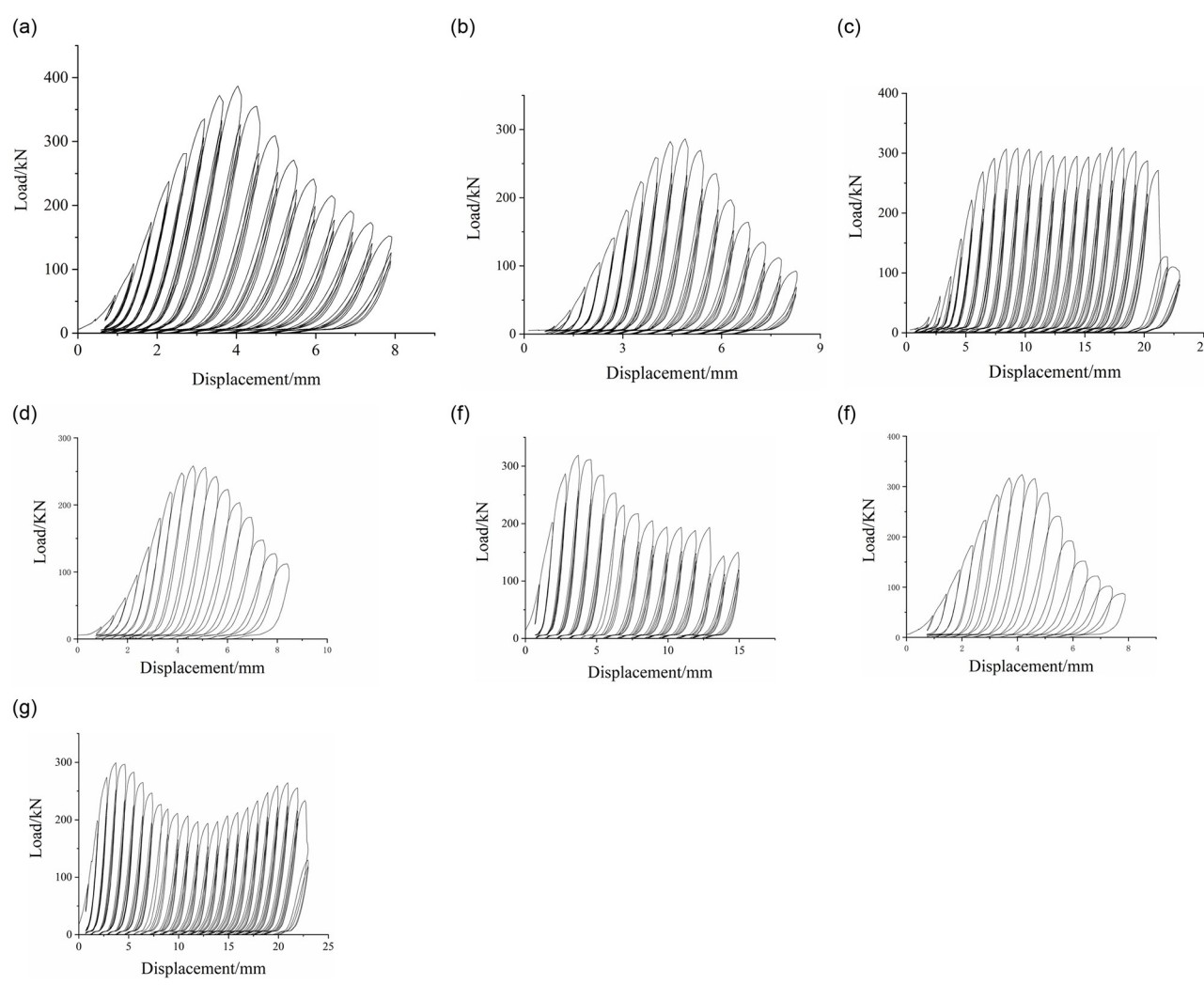

**Fig 7. Load-displacement curves of specimens.** (a)C1W-1. (b)C2W-1. (C)C3S-1. (d)C4W-1. (e)C5S-1. (f)C6W-1. (g) C7S-1.

and initial stiffness are basically not affected. When the concentration of salt solution is 15%, for the unreinforced specimen (C2W-1), the displacement after the load reaches the ultimate bearing capacity is about 4mm, and the specimen begins to yield. With the increase of displacement, the load basically declines linearly. When the displacement is applied to 8mm, the specimen fails. For the SMA reinforced specimen (C3S-1), when the displacement is about 7mm, the load reaches the ultimate bearing capacity and the specimen begins to yield. With the increase of displacement, the load does not decline markedly, but stabilizes in a certain range to form a stable bearing stage; after that, with the increase of displacement, the load shows a slight upward trend and the load decreases slowly. It can be seen that the bearing capacity, stiffness and ductility of reinforced specimens are considerably enhanced compared with those of unreinforced specimens, but the improvement of bearing capacity is obscure.

As can be seen from Fig 7b, 7d and 7f, three different concentrations of salt solution corrosion diminish the bearing capacity of reinforced concrete columns, and with the increase of salt solution corrosion concentration, the bearing capacity gradually declines, and the reduction range is smaller and smaller. With the increase of salt solution corrosion concentration, the ductility and stiffness are basically not notably affected.

As can be seen from Fig 7c, 7e and 7g, with the increase of salt solution concentration, the stiffness and ductility of the test piece gradually decline. The greater the increase of concentration, the smaller the decrease of stiffness and ductility of the test piece. When the ultimate bearing capacity is reached; for the reinforced specimen with 15% concentration, the specimen presents a stable bearing stage, followed by a short strengthening stage; for the reinforced specimen with 20% concentration, there is an obvious falling section, and the whole stress is stepped; For the reinforcement with a concentration of 25%, the whole stress is in a "V" shape. After reaching the bearing capacity, there is an obvious decline stage, and the concrete shows softening characteristics, followed by an obvious rise stage, showing strengthening characteristics. It can be seen that with the increase of salt solution concentration, the axial compression performance of the specimen changes from stable bearing to unstable bearing, and the ultimate bearing capacity declines with the increase of salt solution concentration.

## Bearing capacity analysis

**Unreinforced specimens.** Fig 8 shows the skeleton curve of reinforced concrete columns with different corrosion concentrations. It can be seen that when the concentration of salt solution increases from 0 to 15%, the axial compressive bearing capacity of reinforced concrete columns declines considerably. When the concentration of salt solution increases from 15% to 25%, the bearing capacity declines by 5.9%. It can also be seen intuitively from Table 7 that among the four different salt solution concentrations, 0% has the largest bearing capacity, and the bearing capacity of the other three concentrations has little difference. It can be seen that the bearing capacity of reinforced concrete columns declines considerably after corrosion; as the salt solution concentration increases, the bearing capacity of the specimen gradually declines, but the decline range is obscure.

**Bearing capacity of specimens strengthened with SMA wire.** Fig 9 shows the skeleton curve of SMA reinforced specimens corroded by different concentrations of salt solution. It can be seen that when the displacement is 3mm, the axial compressive load of the three salt solution concentrations reaches the peak nearly at the same time, and the greater the concentration, the smaller the bearing capacity. It can be seen that the bearing capacity of the specimen declines with the increase of the concentration of salt solution. From the concentration curve with the concentration of 15%, it is obvious that the whole curve basically presents a double peak shape and has an obvious bearing capacity strengthening stage, indicating that

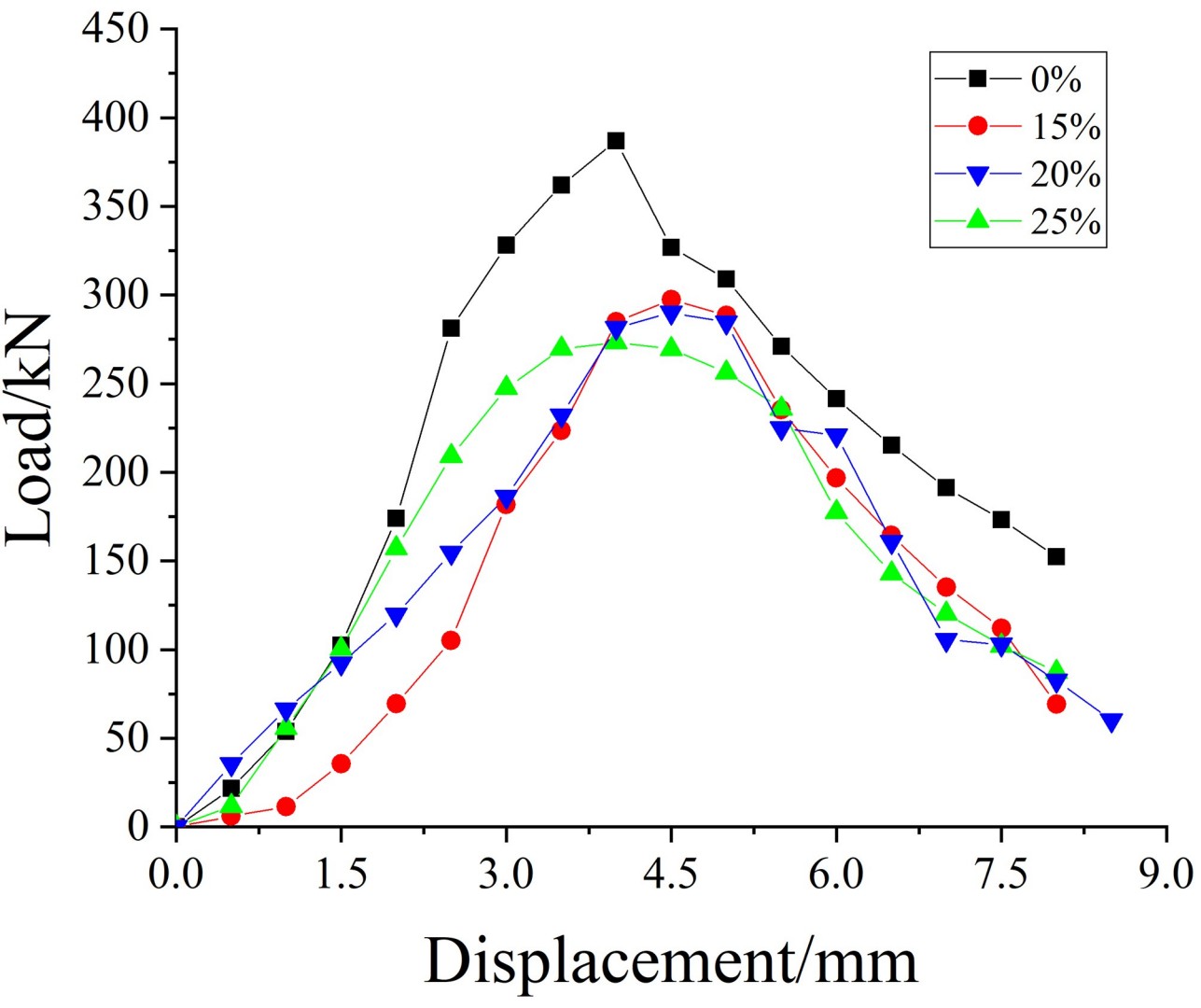

**Fig 8. Skeleton curves of specimens with different concentrations of salt solution.**

SMA has a good reinforcement effect on the specimens with the concentration of 15% and enhanced the bearing capacity. It can also be seen from Table 8 that the reinforcement effect is the best when the concentration is 15%, and the bearing capacity of the concentration of 20% and 25% is reduced by about 4.4% and 11.03% respectively compared with the bearing capacity of the concentration of 15%. There is a small difference between the reinforcement effect of the concentration of 15% and 20% and a large difference between the reinforcement effect of the concentration of 25%, indicating that SMA reinforcement can play a role in a certain concentration of salt solution.

**Table 7. Axial compressive ultimate bearing capacity of specimens with different concentrations of salt solution.**

| Salt solution concentration/% | 0% | 15% | 20% | 25% |
|---|---|---|---|---|
| Ultimate bearing capacity/kN | 387 | 290 | 287 | 273 |

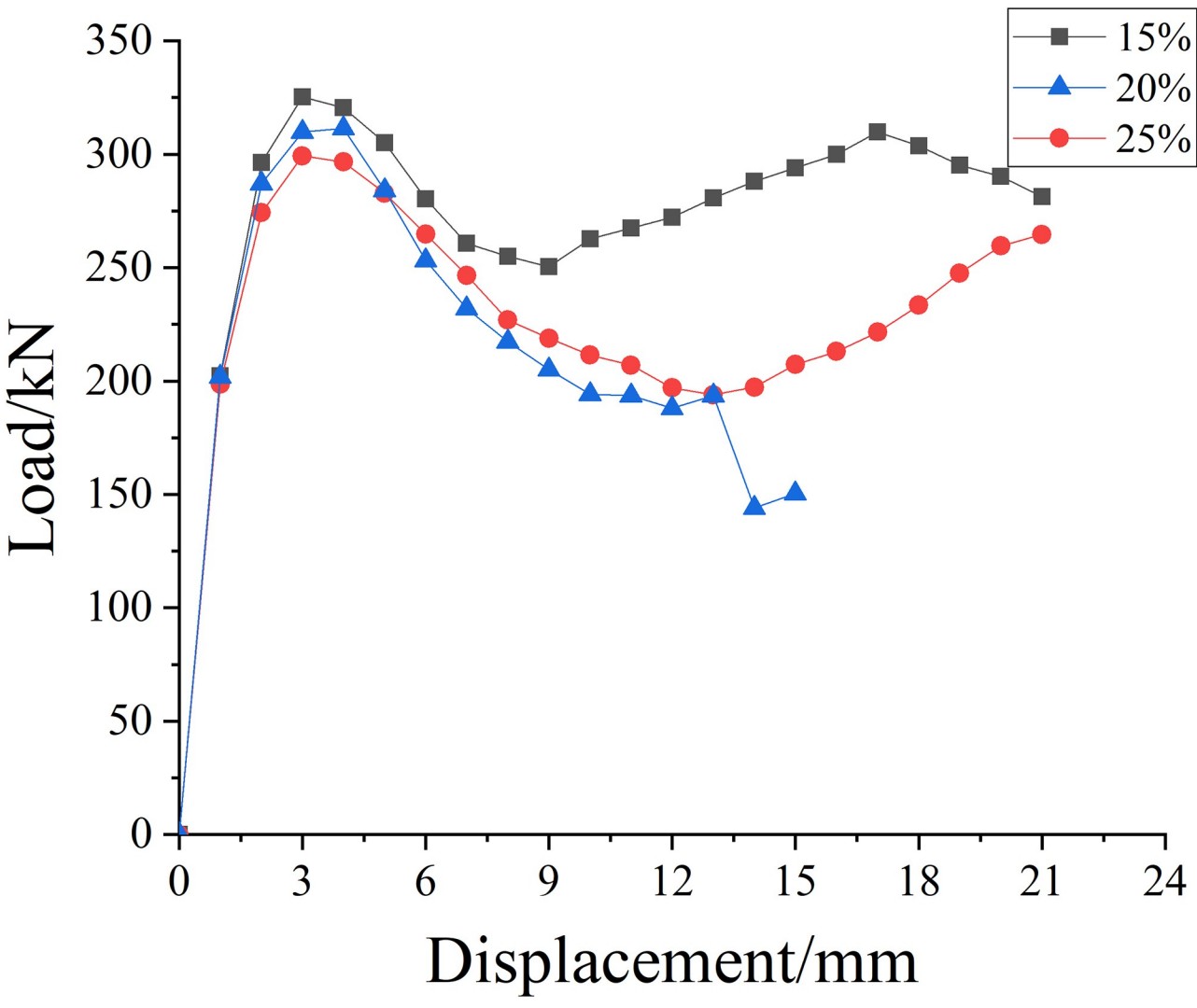

**Fig 9. Skeleton curves of SMA strengthened specimens with different concentrations of salt solution.**

## Energy analysis

**Axial compression energy dissipation performance of unreinforced specimens.** Fig 10 is the variation curve of energy consumption with displacement of the specimen under the corrosion of different concentrations of salt solution. It can be seen from Fig 10 that when the salt solution concentration is 0%, the energy consumption capacity of the specimen is the largest. With the continuous increase of the salt solution concentration, the energy consumption capacity of the specimen declines. When the concentration of salt solution increases from 0% to 15%, the axial pressure energy dissipation capacity of the specimen declines considerably.

**Table 8. Ultimate bearing capacity of specimens strengthened with SMA in different concentrations of salt solution.**

| Salt solution concentration/% | 15% | 20% | 25% |
|---|---|---|---|
| Ultimate bearing capacity/kN | 332 | 318 | 299 |

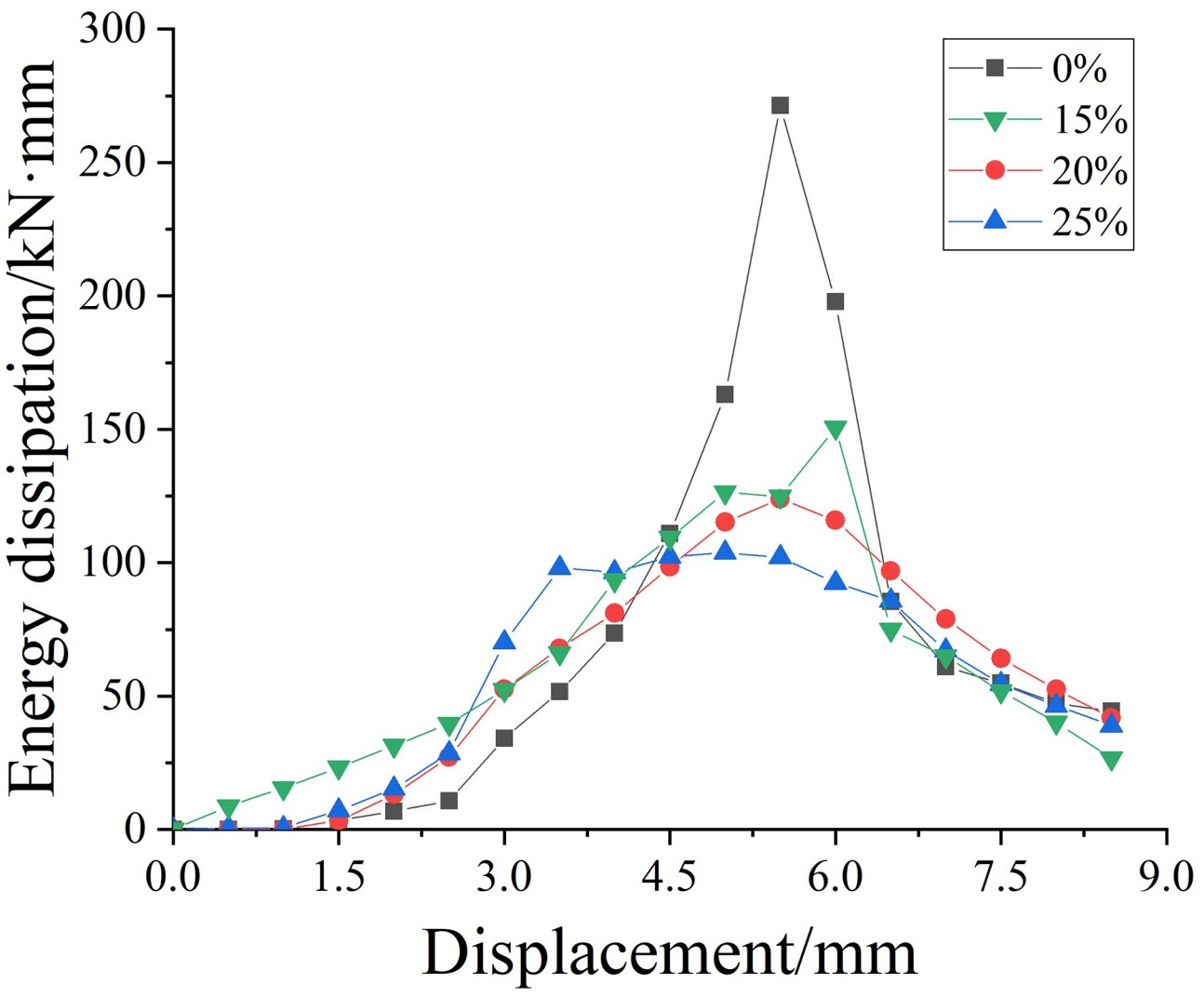

**Fig 10. Energy dissipation displacement curves of specimens with different concentrations of salt solution.**

When the concentration of salt solution increases from 15% to 25%, the energy dissipation capacity does not decline significantly. It can be seen that the energy consumption performance of corroded specimens is far less than that of non-corroded specimens, which declines considerably the energy consumption performance of specimens after corrosion. It can be seen from Table 9 that with the increase of corrosion concentration, the energy consumption performance does not decline significantly, and the total energy consumption capacity of the test piece decreases by about 9.5%, 6.1% and 2.2% respectively, compared with the previous concentration.

**Energy dissipation behavior of specimens strengthened with SMA wire under axial compression.** Fig 11 is the energy consumption versus displacement curve of SMA

**Table 9. Total axial energy consumption of unreinforced specimens with different concentrations of salt solution.**

| Salt solution concentration/% | 0% | 15% | 20% | 25% |
|---|---|---|---|---|
| Axial compression energy consumption value/(kN·mm) | 1216 | 1100 | 1033 | 1010 |

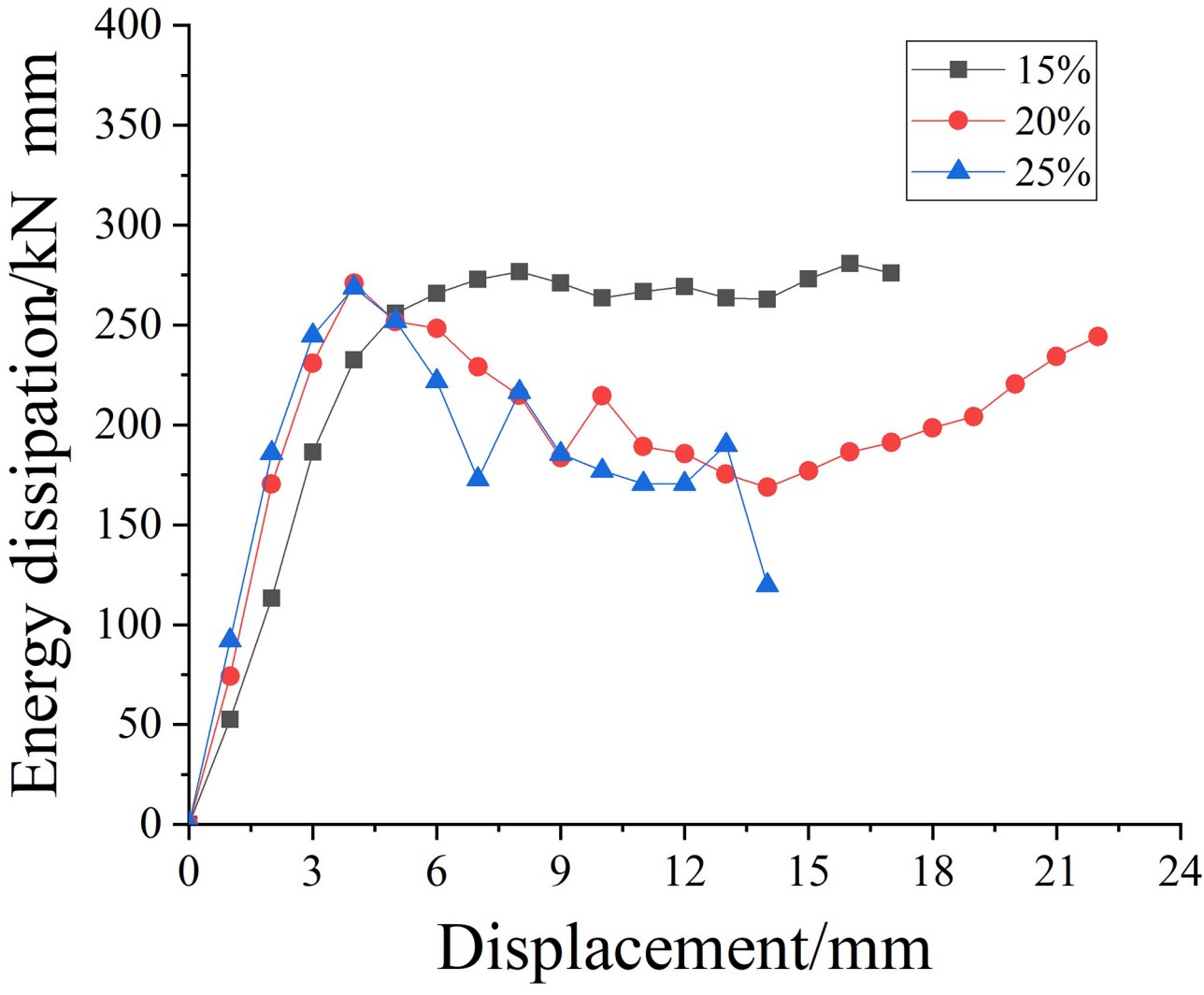

**Fig 11. Energy dissipation displacement curves of specimens strengthened with SMA in different concentrations of salt solution.**

reinforced specimens with different concentrations of salt solution. It can be seen that the energy consumption rate increases with the increase of salt solution concentration; when the energy consumption value exceeds its limit value, the energy consumption capacity of SMA reinforced specimens with salt solution concentration of 20% and 25% decreases to a certain extent, while the energy consumption capacity with salt solution concentration of 15% remains stable.

It can be seen that with the increase of salt solution concentration, the energy consumption performance develops from stable energy consumption to unstable energy consumption, and the energy consumption performance is declining. It can be seen from Table 10 that when the

**Table 10. Total energy consumption of specimens strengthened with SMA in different concentrations of salt solution under axial compression.**

| Salt solution concentration/% | 15% | 20% | 25% |
|---|---|---|---|
| Axial compression energy consumption value/(kN·mm) | 4747 | 4354 | 2668 |

concentration of salt solution is 15% and 20%, the difference of total energy consumption effect of SMA wire reinforced specimen is very small, about 8.3%; when the concentration of salt solution is 25%, the total axial compressive energy dissipation capacity of strengthened specimens decreases by about 38.7% compared with the previous concentration, indicating that the axial compressive energy dissipation capacity of SMA strengthened specimens decreases more and more with the increase of salt solution concentration. From the comparison between Tables 9 and 10, the energy dissipation performance of SMA reinforced specimens with a concentration of 15% is about 4 times higher than that of non- corroded and unreinforced specimens, indicating that SMA reinforcement improves the energy dissipation performance of specimens.

### Residual deformation

**Unreinforced specimens.**   Fig 12 shows the residual deformation curve of reinforced concrete columns under different corrosion concentrations. It can be seen from the comparison diagram of residual deformation of salt solution concentration of 0%, 15%, 20% and 25% and Table 11 that the residual deformation of reinforced concrete column with salt solution concentration of 25% is the largest. With the increase of salt solution concentration, the total residual deformation of reinforced concrete columns increases. Within the displacement loading level of 4.5mm, the residual displacement of the four specimens is not very different. When the loading displacement level exceeds 4.5mm, the residual deformation of reinforced concrete columns with salt solution concentration of 15%, 20% and 25% gradually increases, indicating that the corrosion of salt solution will make reinforced concrete columns lose certain self recovery ability.

**SMA reinforced specimen.**   Fig 13 shows the residual deformation curve of reinforced concrete columns under different corrosion concentrations. It can be seen from Fig 13 and Table 12 that with the increase of salt solution concentration, the residual deformation of reinforced concrete columns strengthened with NiTi SMA wire is almost the same before the loading displacement level is 3mm. When the loading displacement exceeds 3mm, the residual deformation of the reinforced column gradually increases with the increase of the concentration of the salt solution, with an increase range of 0.18%. Nevertheless, when the concentration of the salt solution is 25%, the SMA wire breaks early due to the large prestress control of the SMA wire, and its deformation value is far less than that of the concentration of 15% and 20%.

## Finite element analysis

### Establishment of finite element model

According to the actual situation of the test, the component model with the same size as the test specimen is established by using ABAQUS software. The concrete damage plastic model is used in the concrete constitutive model, which has good convergence; the reinforcement material model adopts the double broken line model. The elastic modulus is taken as $2.07 \times 10^5$MPa, Poisson's ratio is 0.3; the SMA constitutive model has a special heuristic SMA constitutive model in the finite element software, so it can be simulated by directly calling the constitutive method of SMA material in the software. This material is elastic before leaving work, so it meets Hooke's law of stress. The elastic modulus is 114000MPa and the poisson ratio is 0.3. The concrete column is simulated by three-dimensional deformable solid element C3D8R, and the reinforcement cage is simulated by wire element T3D2. SMA wire is imported into ABAQUS software to form spiral SMA wire components after the solid model is established in CAD software, and C3d8i unit is used for simulation.

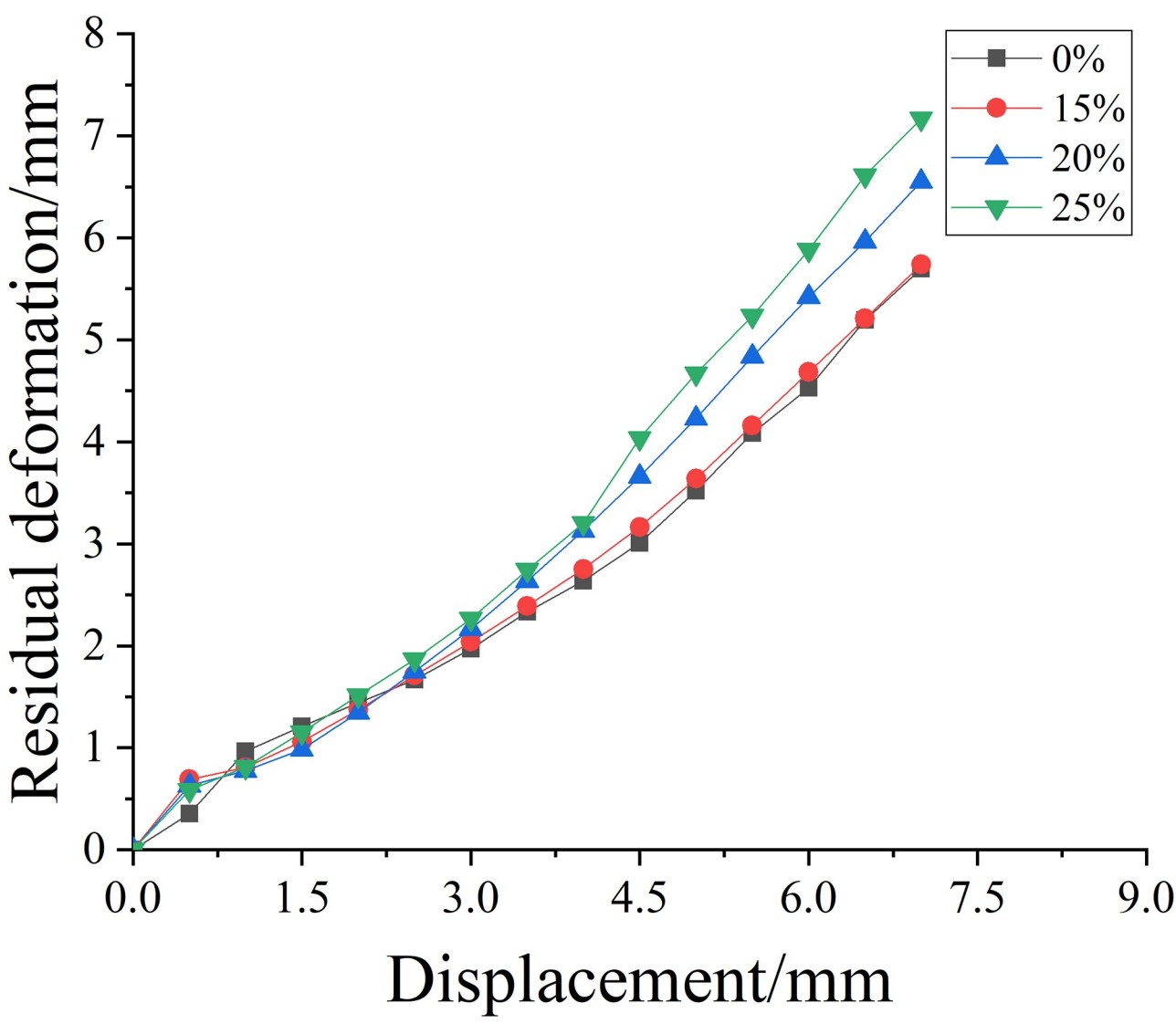

**Fig 12. Residual deformation curves of reinforced concrete columns with different salt solution concentrations.**

The connection between the reinforcement cage and the concrete unit is realized by embedded constraints; the constraint between SMA wire and concrete is defined as tie. Concrete is the primary surface area and SMA wire is the slave surface area. For the hinge constraint of the degree of freedom at the bottom end of the reinforced concrete column, a datum point RP-1 is created at the top of the reinforced concrete column, and the datum point is coupled with the top surface of the reinforced concrete column to constrain two horizontal degrees of freedom perpendicular to the normal direction of the top surface of the concrete column. Each part of the model is shown in Fig 14.

**Table 11. Total residual deformation of reinforced concrete columns with different salt solution concentrations under axial compression.**

| Salt solution concentration/% | 0% | 15% | 20% | 25% |
|---|---|---|---|---|
| Total residual value/mm | 6.896 | 6.899 | 7.140 | 7.168 |

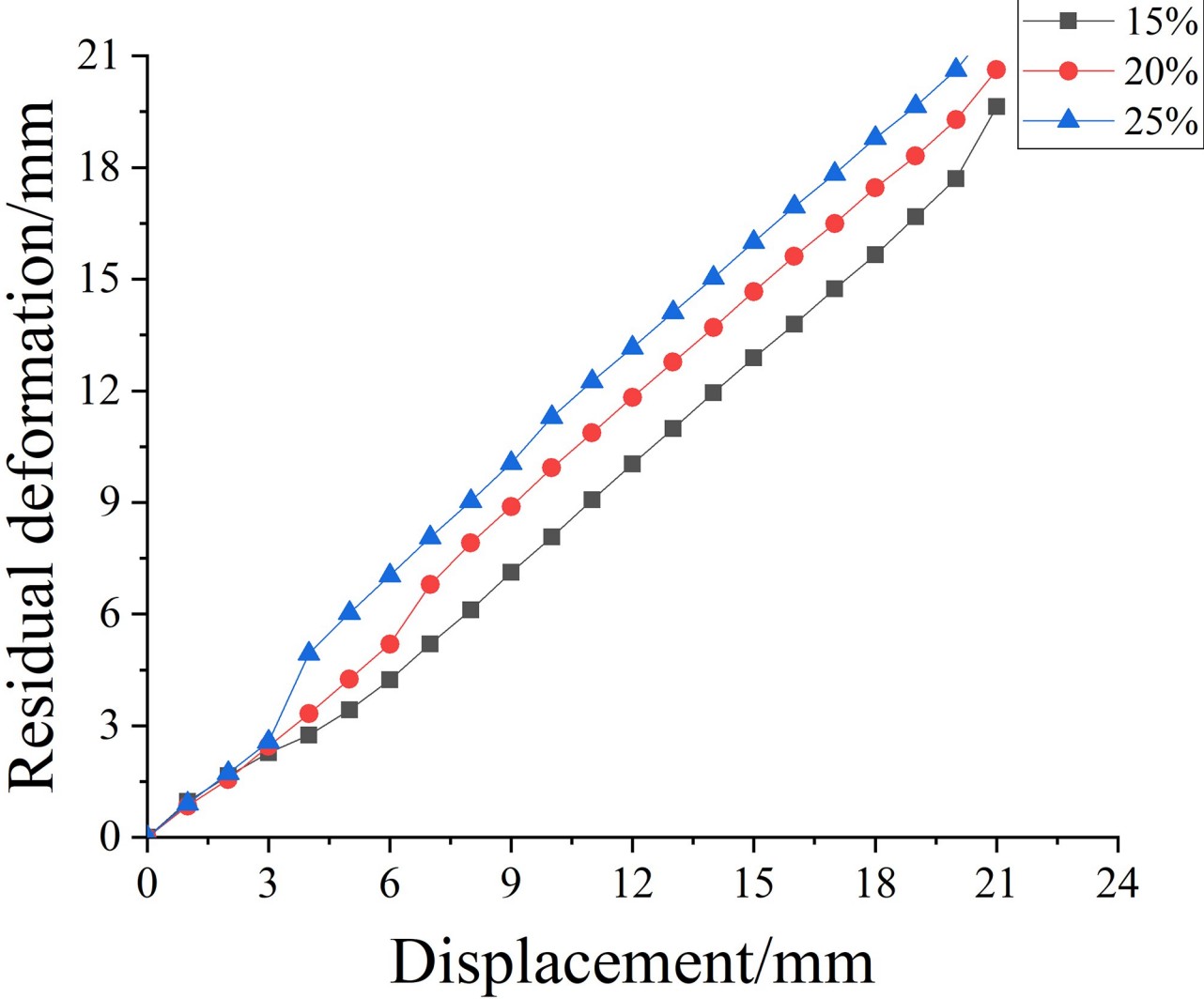

**Fig 13. Residual deformation curves of reinforced concrete columns strengthened with SMA with different salt solution concentrations.**

## Verification of finite element model

**Failure mode verification.** The failure modes of the finite element simulation and test of the specimen are shown in Fig 15. Fig 15a and 15b show the comparison between test and numerical simulation of unreinforced specimens. It can be seen from the Fig 15a and 15b that the failure modes of the numerical simulation and the test are basically the same. The deformation in the middle of the concrete is massive, and there is a lantern shape, which is consistent with the deformation position of the specimen during the test loading. Micro cracks first appear in the test, and then massive deformation.

**Table 12. Axial total residual deformation of reinforced concrete columns strengthened by SMA with different salt solution concentrations.**

| Salt solution concentration/% | 15% | 20% | 25% |
|---|---|---|---|
| Total residual value/mm | 20.627 | 20.665 | 21.961 |

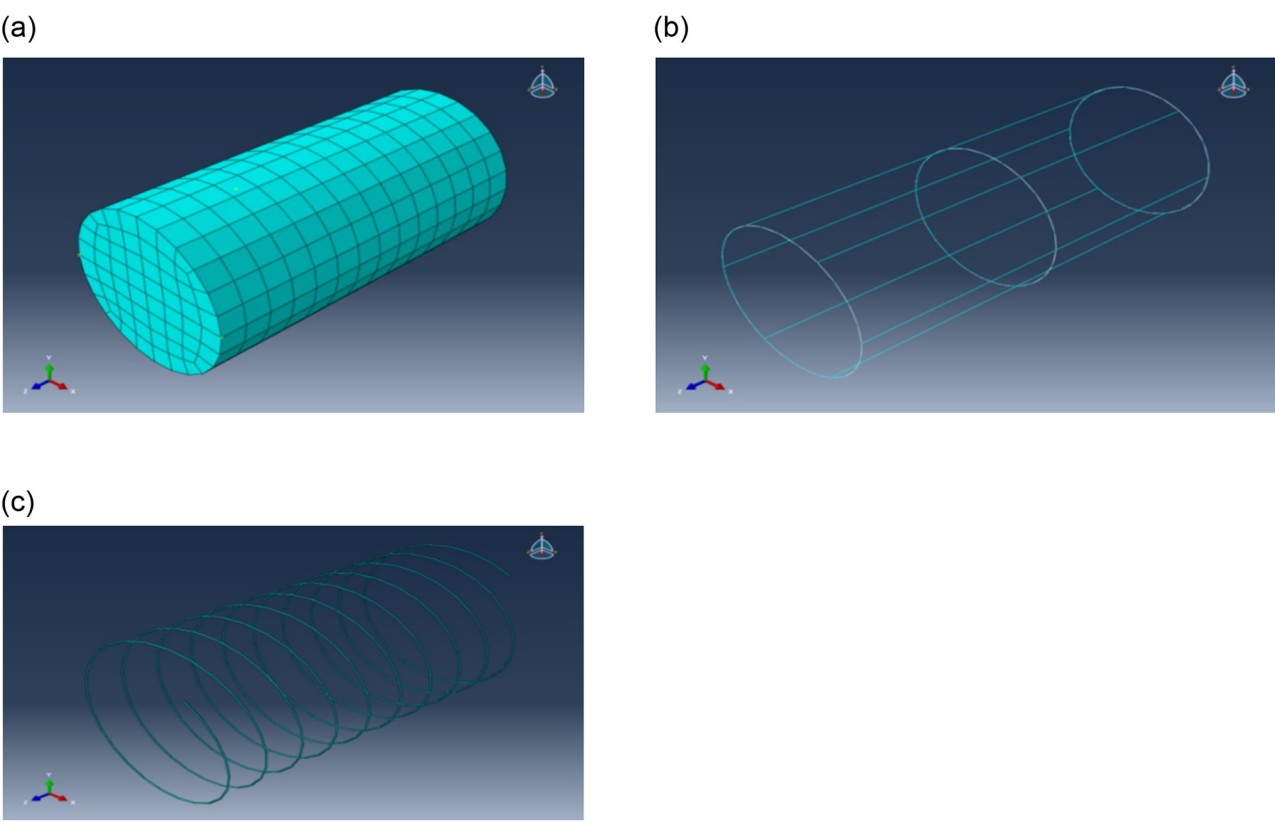

**Fig 14. Model diagram of each part.** (a) Concrete. (b) Reinforcement cage. (c) SMA wire.

It can be seen from Fig 15c and 15d that the deformation of the SMA wire reinforced specimen is concentrated in the upper part of the column, and the deformation is small, there are small cracks, there is no obvious swelling, and there is a good deformation capacity, and the test is basically consistent with the numerical simulation deformation.

**Bearing capacity verification.** Through the finite element modeling and analysis of SMA reinforced specimen and unreinforced specimen, the load displacement curve simulation results are compared with the test results, as shown in Fig 16. The test value and the numerical simulation value are comparatively close on the whole, which verifies the applicability and accuracy of the model, sufficiently reflects the mechanical performance of the reinforced concrete column strengthened with super elastic SMA wire under uniaxial compression, and verifies the accuracy of the test results. From the figure, it can be seen that the bearing capacity of the corroded concrete column after being strengthened with SMA wire is considerably enhanced, and compared with the unreinforced specimen, the specimen strengthened with SMA wire has a stable bearing capacity. The numerical simulation curve of the specimen is basically consistent with the test value curve before reaching the ultimate bearing capacity. From the numerical analysis, the simulation value of the specimen's ultimate bearing capacity is similar to the test value of the specimen's ultimate bearing capacity, but it is somewhat higher than the test value, and the error between the peak deformation value and the test peak deformation value is somewhat larger. This is because the reinforced concrete columns are manually poured during the pouring process, which will inevitably lead to accidental errors, and uneven mixing may occur. Result in a small number of weak areas in the test piece. In the

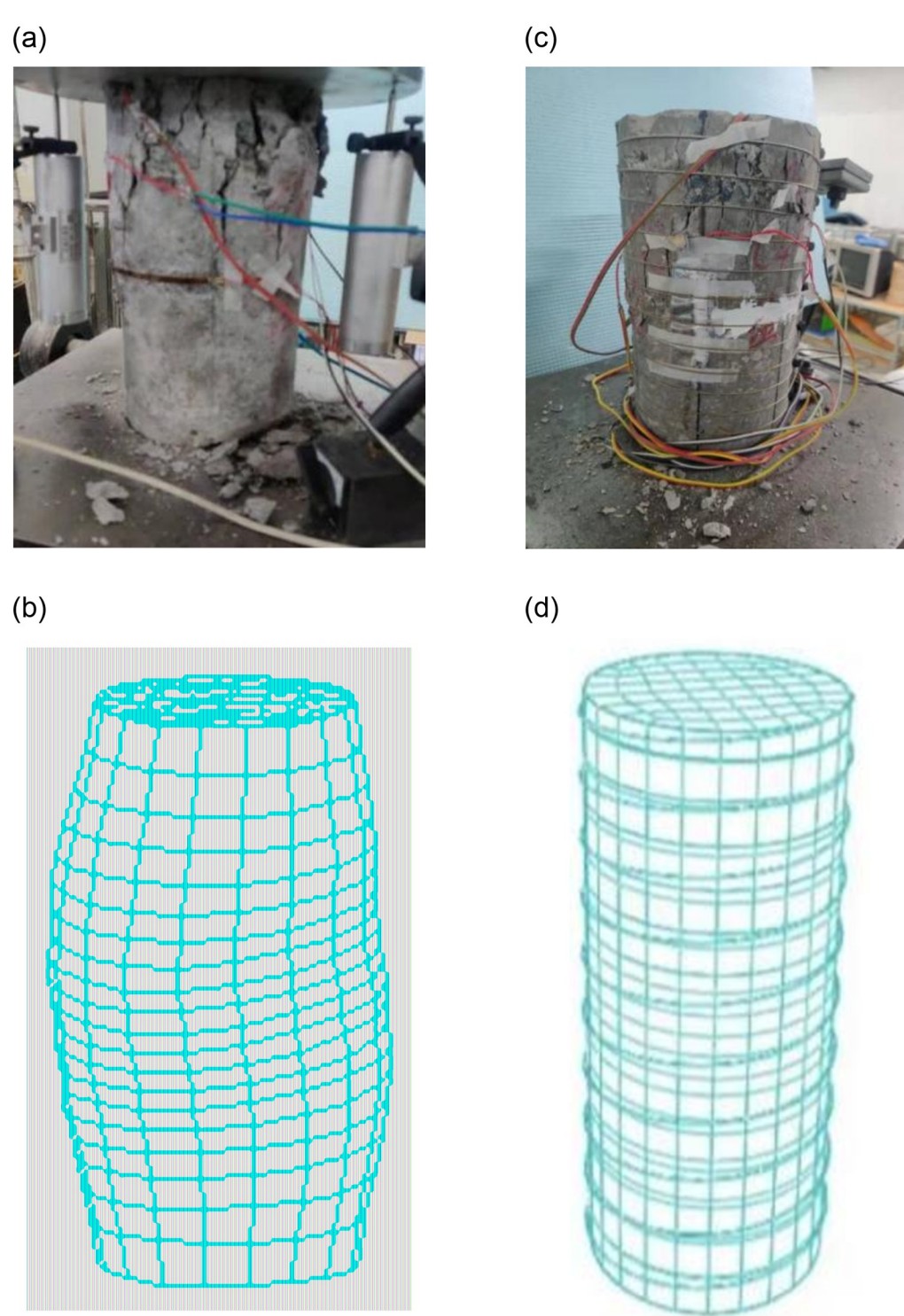

**Fig 15. Comparison of specimen failure and deformation.** (a)Test failure deformation. (b) Numerical simulation of deformation. (c)Test failure deformation. (d) Numerical simulation of deformation.

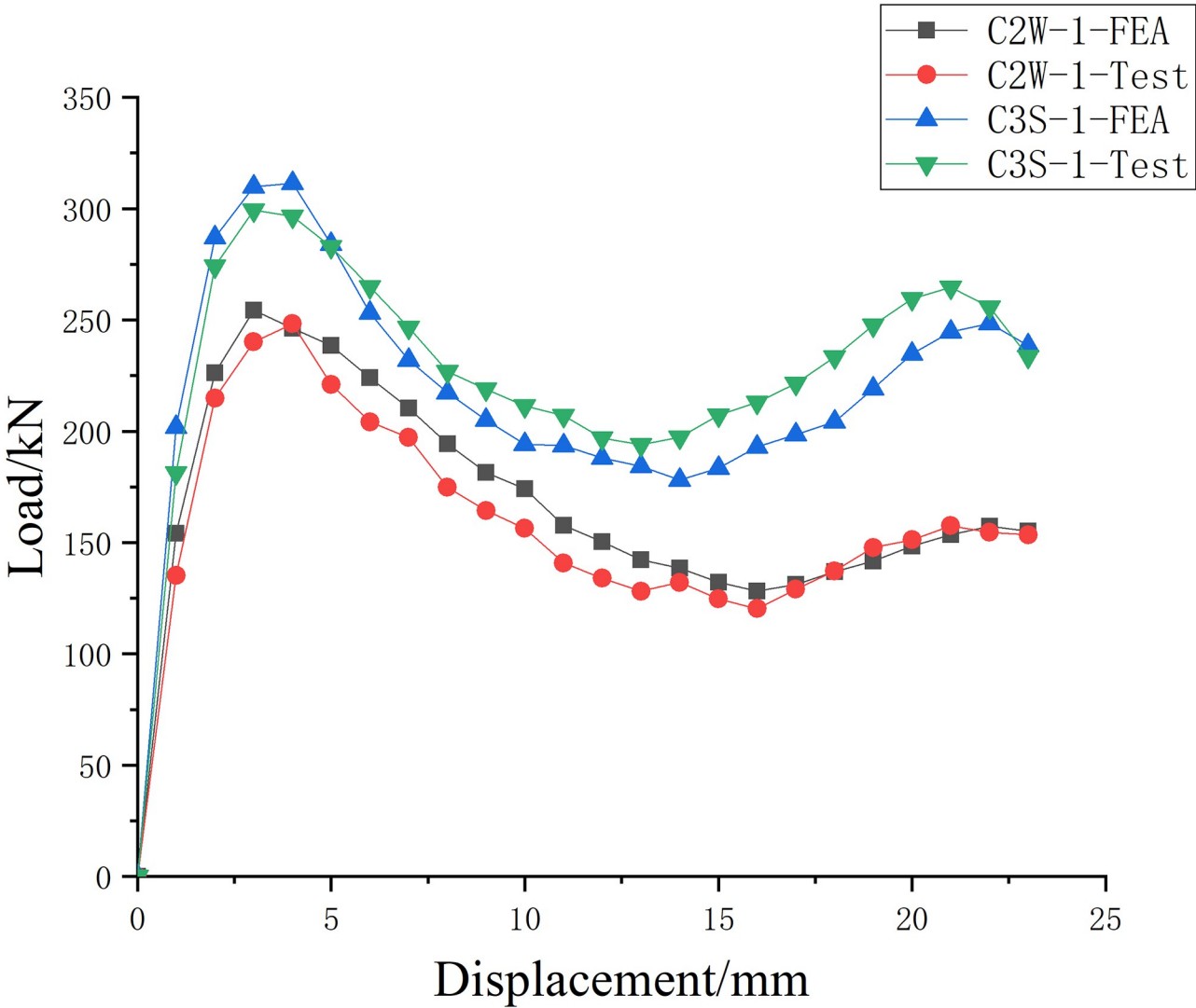

**Fig 16. Load displacement curve.**

process of numerical simulation, it is assumed that the material is the ideal state of isotropy, and the stress and deformation are better than the actual situation.

## Extended parameter analysis

Based on the finite element model, the effects of SMA wire spacing, SMA wire diameter, concrete strength and cover thickness on the bearing capacity of reinforced concrete short columns are analyzed.

**SMA wire spacing.** In order to study the influence of SMA wire reinforcement spacing on reinforced concrete columns, five different SMA wire reinforcement spacing are designed, which are 2.5mm, 5mm, 10mm, 20mm and 40mm, respectively. Other factors remain unchanged, that is, the concrete strength grade is C25 and the SMA wire diameter is 1.2mm. The load displacement curves of specimens with various SMA wire reinforcement spacing are calculated, and the results are shown in Fig 17.

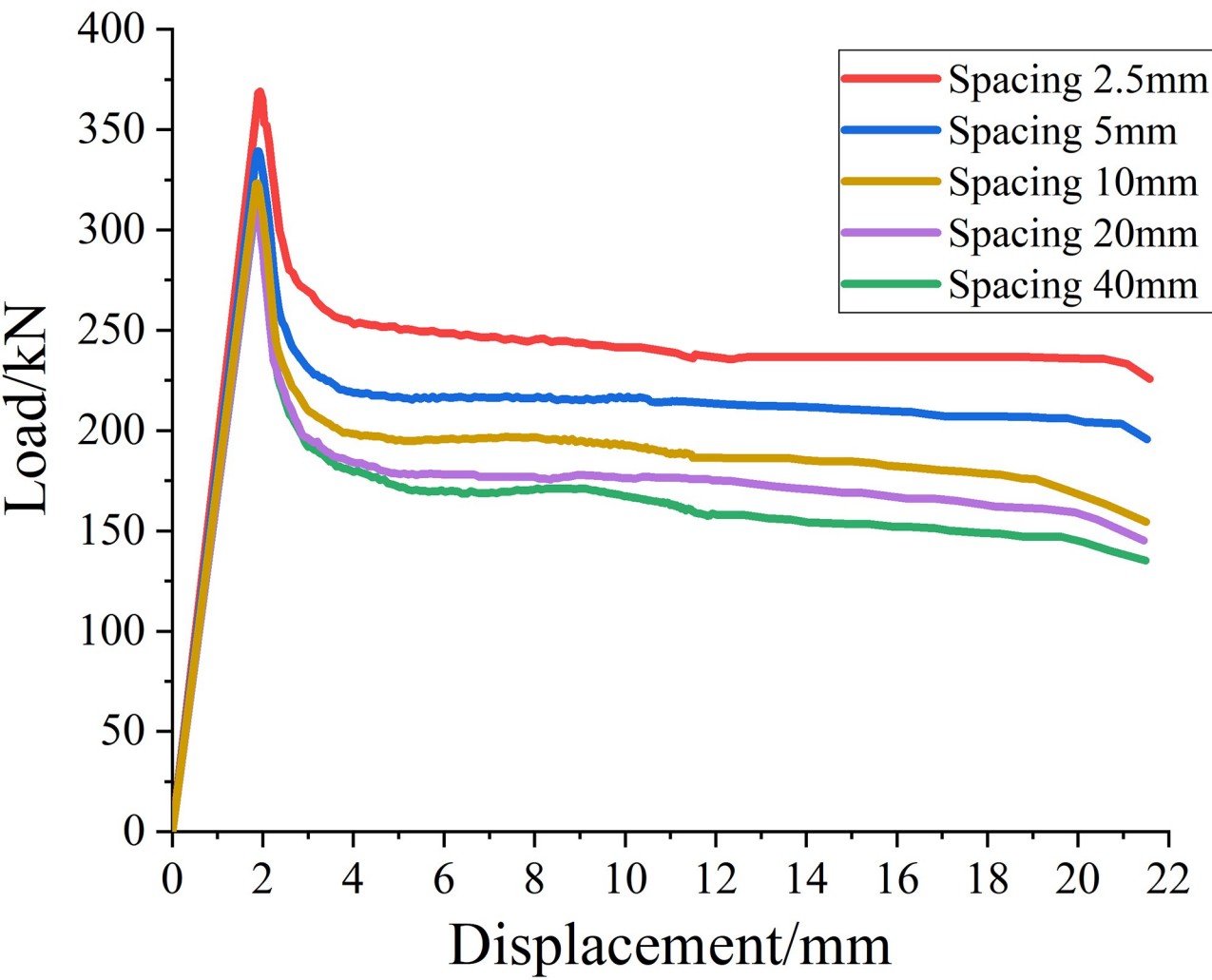

**Fig 17. Load-shift curve of specimens with different reinforcement spacing of SMA wires.**

It can be seen from Table 13 and Fig 17 that when other factors remain unchanged, with the increase of the reinforcement spacing of hyperelastic SMA wire (equivalent to the decrease of reinforcement amount), the ultimate load value of reinforced concrete column decreases continuously, and its ultimate bearing capacity relative to the upper reinforcement spacing decreases by about 8.1%, 4.7%, 2.4% and 1.8% respectively.

**SMA wire diameter.** In order to study the influence of SMA wire diameter on reinforced concrete columns, five different SMA wire diameters are designed for reinforcement, which are 0.6mm, 1.2mm, 2.4mm, 3.6mm and 7.2mm, respectively. Other factors remain unchanged, that is, the concrete strength grade is C25. The load displacement curves of reinforced specimens with different SMA wire diameters are calculated, and the results are shown in Fig 18.

**Table 13. Ultimate load values of specimens with different reinforcement spacing of SMA.**

| SMA wire reinforcement spacing/mm | 2.5 | 5 | 10 | 20 | 40 |
|---|---|---|---|---|---|
| Ultimate load value/kN | 369.08 | 339.27 | 323.39 | 315.48 | 313.68 |

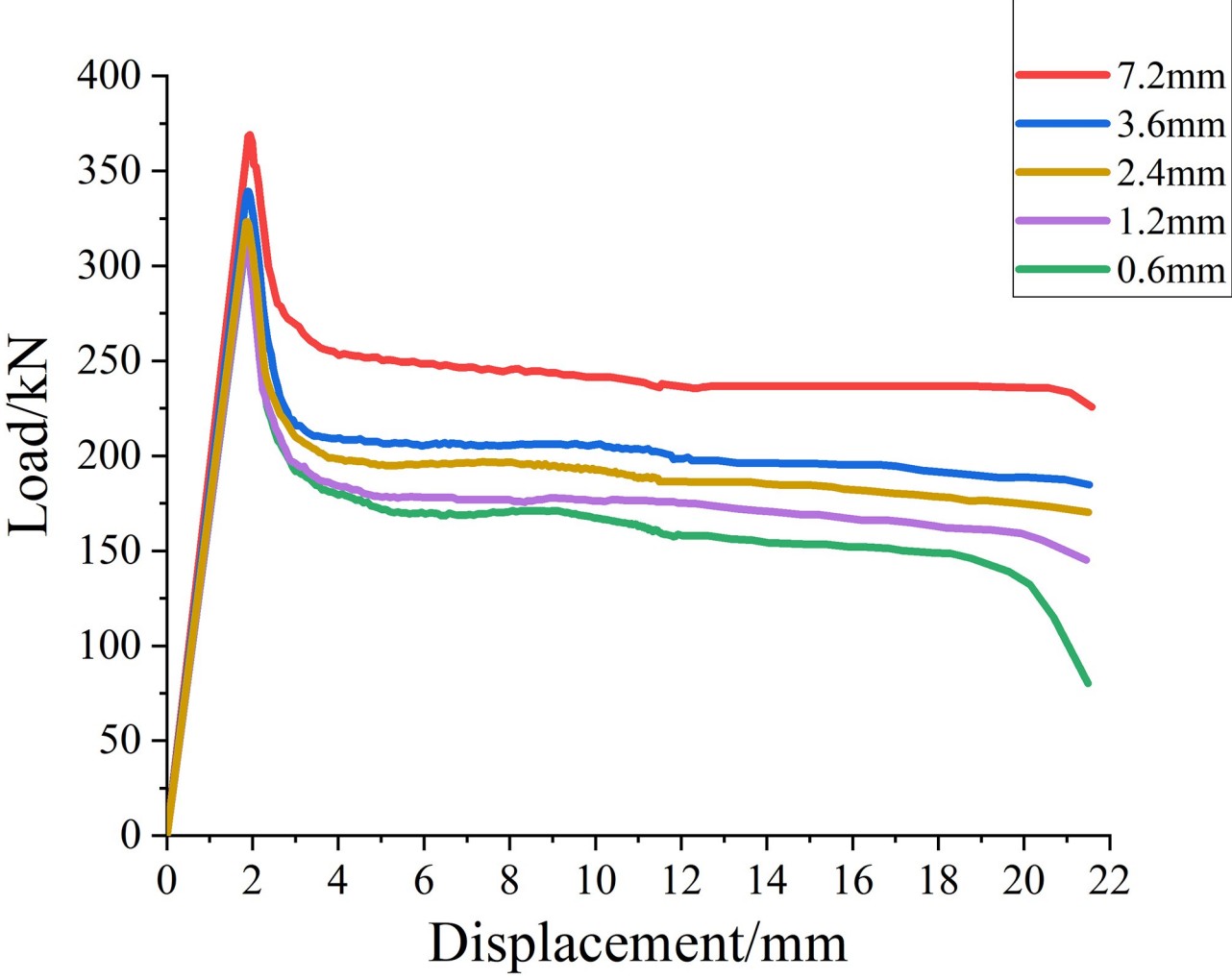

**Fig 18. Load-displacement curves of reinforcement parts with different diameter of SMA wires.**

It can be seen from Table 14 and Fig 18 that when other factors remain unchanged, with the increase of the diameter of hyperelastic SMA wire (equivalent to the increase of reinforcement), the ultimate load value of reinforced concrete column continues to rise, and its ultimate bearing capacity relative to the upper reinforcement spacing increases by about 1.3%, 2.2%, 3% and 8.1% respectively.

**Concrete strength.** In order to study the influence of concrete strength on reinforced concrete columns, five concrete columns with different strength grades are designed for reinforcement, which are C25, C30, C35, C40, and C45, respectively. Other factors remain unchanged, that is, the concrete reinforcement spacing is 20mm and the diameter of SMA wire is 1.2mm. The load displacement curves of specimens with different concrete strength strengthened with different SMA wires are calculated. The results are shown in Fig 19.

**Table 14. Ultimate load values of reinforcement parts with different diameter of SMA wires.**

| SMA wire diameter/mm | 0.6 | 1.2 | 2.4 | 3.6 | 7.2 |
|---|---|---|---|---|---|
| Ultimate load value/kN | 307.29 | 314.14 | 318.33 | 327.95 | 354.44 |

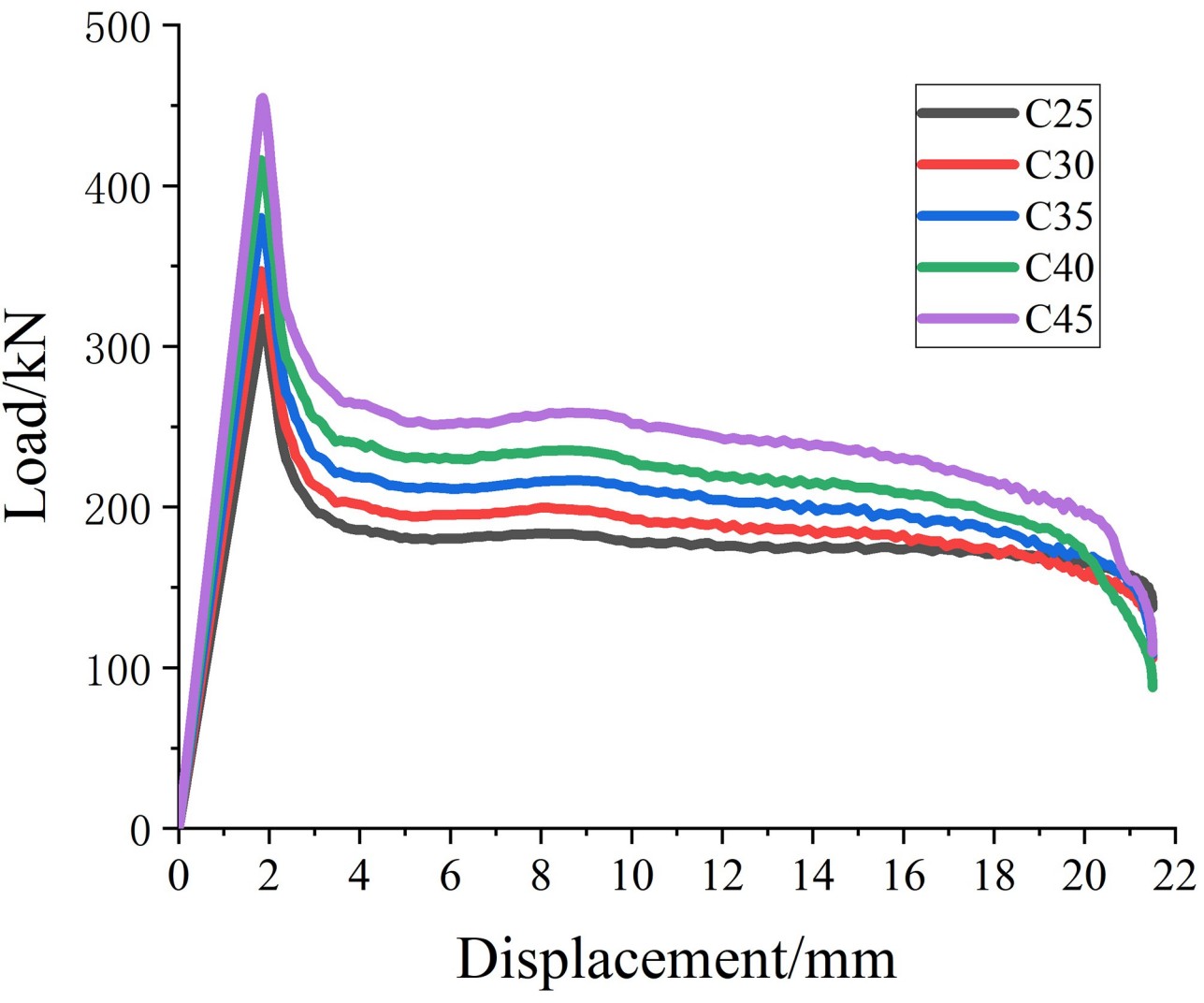

**Fig 19. Load—Displacement curve of reinforcement parts with different concrete strength.**

It can be seen from Table 15 and Fig 19 that when other factors remain unchanged, the ultimate load value of reinforced concrete column increases continuously with the increase of concrete strength, and its ultimate bearing capacity relative to the upper reinforcement spacing increases by about 9.4%, 9.5%, 9.5% and 9%, respectively.

**Thickness of concrete cover.** In order to study the influence of cover thickness on reinforced concrete columns, five concrete columns with different cover thickness are designed for reinforcement, which are 5mm, 20mm, 25mm, 30mm and 35mm, respectively. Other factors remain unchanged, that is, the concrete reinforcement spacing is 20mm and the diameter of SMA wire diameter is 1.2mm. The load displacement curves of specimens with different cover thickness strengthened by SMA wire are calculated. The results are shown in Fig 20.

**Table 15. Ultimate load values of reinforcement parts with different diameter of SMA wires.**

| Concrete strength grade | C25 | C30 | C35 | C40 | C45 |
|---|---|---|---|---|---|
| Ultimate load value/kN | 317.14 | 347.06 | 379.97 | 415.97 | 453.57 |

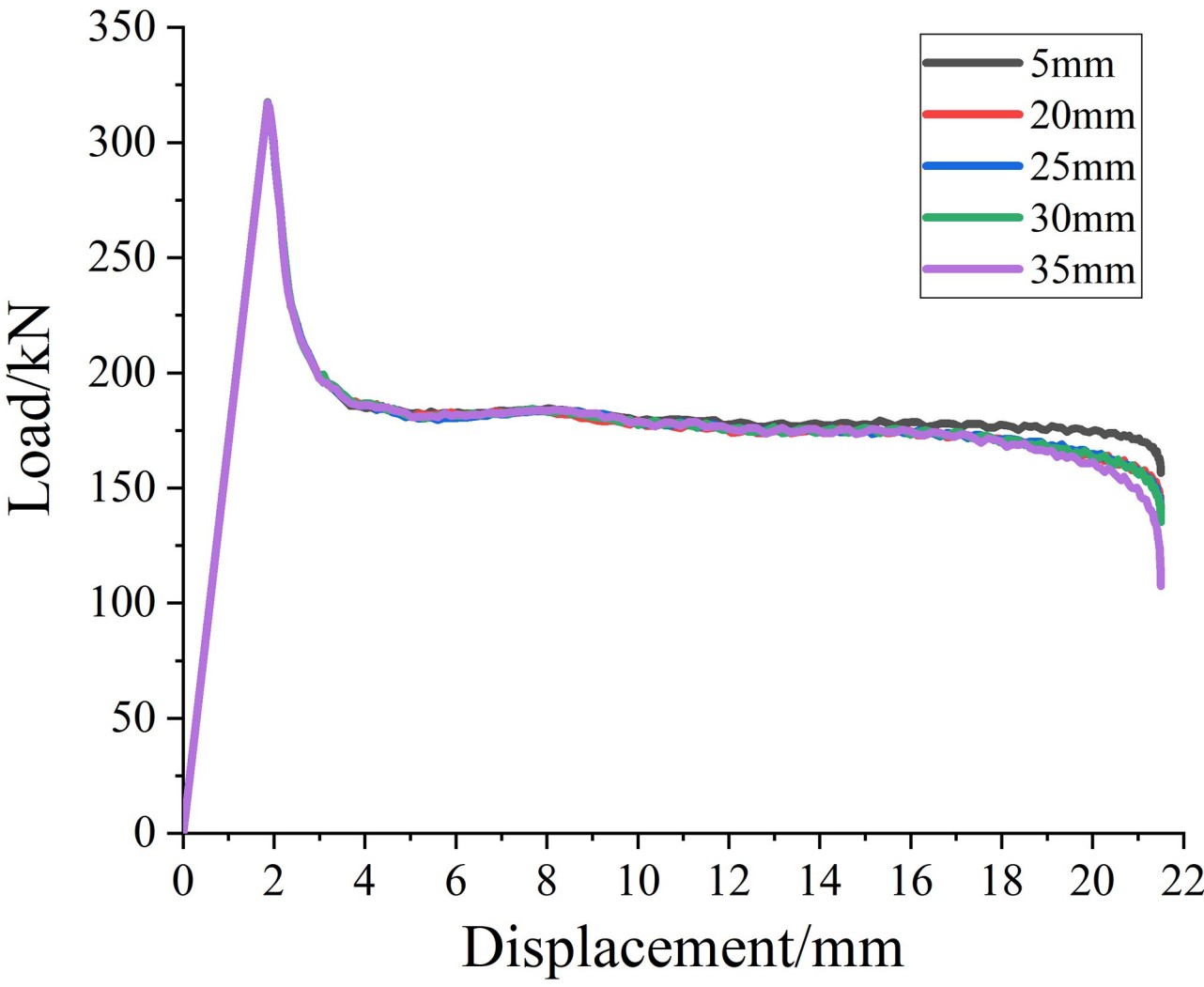

**Fig 20. Load—Displacement curve of reinforcement parts with different thickiness of concrete cover.**

It can be seen from Table 16 and Fig 20 that when other factors remain unchanged, the ultimate load value of reinforced concrete column changes little with the increase of protective layer thickness, and its ultimate bearing capacity enhances by about 0.8%, 0.3%, 0.003% and 0.003% respectively, relative to the upper reinforcement spacing.

## Conclusions and discussions

In this paper, experimental and numerical studies of reinforced concrete columns are carried out in the background of offshore engineering. After the concrete columns were treated with salt solution corrosion, the concrete columns were strengthened with SMA wire, and the monotonic loading test was performed. Finally, the experimental results are highly consistent

**Table 16. Ultimate load value of reinforcement with different thickness of concrete cover.**

| Thickness of concrete cover/mm | 5 | 20 | 25 | 30 | 35 |
|---|---|---|---|---|---|
| Ultimate load value/kN | 317.51 | 317.25 | 317.14 | 317.13 | 317.12 |

with the finite element results, which verifies the validity of the finite element model. The main conclusions and discussions are as follows:

1. The initial stiffness and bearing capacity of the unreinforced specimens corroded by seawater are lower than those of the uncorroded and unreinforced specimens, but the ductility has no effect. The failure mode of corroded and unreinforced specimens is local cracks, which are bulging in the middle of the column, forming local through cracks, accompanied by internal concrete crushing failure.

2. Compared with unreinforced specimens, SMA reinforced specimens have stable bearing stage and strengthening stage, which enhances the bearing capacity of specimens and prolongs the failure time of specimens. When the concentration of salt solution is 15%, the mechanical properties of the specimens strengthened with SMA wire are considerably enhanced.

3. The corrosion of salt solution will decline the bearing capacity, energy dissipation capacity, stiffness and ductility of specimens, but the impact is small.

4. Compared with unreinforced specimens, SMA reinforced specimens can greatly improve the energy dissipation performance of specimens. When the salt solution concentration is 15%, the energy dissipation performance of the specimens strengthened with SMA wire is about 4 times higher than that of the specimens without reinforcement. With the increase of salt solution concentration, the increase of axial compression energy dissipation performance of the specimens strengthened with SMA becomes smaller and smaller.

5. The corrosion of salt solution will increase the residual deformation and weaken the deformation ability. Nevertheless, SMA wire reinforcement can decline the residual deformation of concrete columns to a certain extent and enhance the deformation capacity. Compared with the unreinforced columns, the residual deformation of SMA wire reinforced specimens is declined by 11.4%.

6. By establishing the finite element model, the accuracy of the model is verified, and the effects of SMA wire spacing, SMA wire diameter, concrete strength and protective layer thickness on the bearing capacity of concrete columns are studied. Among them, The strength grade of concrete is the most important factor affecting the bearing capacity of the column, which is enhanced by up to 9.5%, while the thickness of the concrete cover is the factors of influencing lowest affecting the bearing capacity of the column, which is only enhanced by 0.8%.

7. First, it is worth noting that the test in this paper is carried out at room temperature. Ni-Ti SMA wire can work stably in a relatively stable room temperature. However, Ni-Ti SMA is extremely sensitive to temperature, which may make it unsuitable for outdoor applications. In order to solve this problem, experimental or numerical simulation research on the influence of temperature sensitivity on the test results should be carried out in the future. Secondly, the above conclusions are obtained through experiments and numerical simulation, but there is no simple formula to calculate the bearing capacity of columns by SMA wire. In order to facilitate practical engineering application in the future, on this basis, a simple formula for bearing capacity calculation of columns by SMA wire should be derived.

## Acknowledgments

The authors gratefully acknowledge the support from Su Liang from Jilin Guanteng Automation Technology Co., Ltd, during the mechanical tests of SMA reinforcement parameters

conducted for this study. And we also wish to thank the anonymous reviewers for their careful evaluations and insightful comments that helped improve the paper.

## Author Contributions

**Conceptualization:** Yu Ding, Yun Guo.

**Data curation:** Bangwen Cai.

**Funding acquisition:** Qiang Pei.

**Investigation:** Zhicheng Xue.

**Methodology:** Qiang Pei.

**Project administration:** Qiang Pei.

**Validation:** Di Cui.

**Writing – original draft:** Qiang Pei, Bangwen Cai.

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
