## [Decision Letter · Decision Letter 0]

13 Jun 2022

PONE-D-22-13623Study on mechanical properties of reinforced concrete columns corroded in simulated seawater strengthened by shape memory alloy wiresPLOS ONE

Dear Dr. Pei,

Thank you for submitting your manuscript to PLOS ONE. After careful consideration, we feel that it has merit but does not fully meet PLOS ONE’s publication criteria as it currently stands. Therefore, we invite you to submit a revised version of the manuscript that addresses the points raised during the review process.

We look forward to receiving your revised manuscript.

Kind regards,

Cheng Fang

Academic Editor

PLOS ONE

Journal Requirements:

"This research was supported by the National Natural Science Foundation of China (Grant Nos. 51878108, 52032011), Department of Science & Technology Guidance Plan Foundation of Liaoning Province (Grant Nos. 2019JH8/10100091), Scientific Research Project of Liaoning Provincial Department of Education (Grant Nos. LJKZ1177). The authors also wish to thank the anonymous reviewers for their careful evaluations and insightful comments that helped improve the paper."

Additional Editor Comments (if provided):

Reviewer #3 recommends Reject for the paper. The authors should carefully respond to the reviewers' comments, otherwise the editor would have no choice but reject the paper.

Reviewers' comments:

Reviewer's Responses to Questions

**Comments to the Author**

1. Is the manuscript technically sound, and do the data support the conclusions?

Reviewer #1: Yes

Reviewer #2: Yes

Reviewer #3: No

2. Has the statistical analysis been performed appropriately and rigorously? 

Reviewer #1: Yes

Reviewer #2: Yes

Reviewer #3: I Don't Know

3. Have the authors made all data underlying the findings in their manuscript fully available?

Reviewer #1: Yes

Reviewer #2: Yes

Reviewer #3: No

4. Is the manuscript presented in an intelligible fashion and written in standard English?

Reviewer #1: Yes

Reviewer #2: Yes

Reviewer #3: No

5. Review Comments to the Author

Reviewer #1: This paper inverstigates mechanical properties of reinforced concrete columns corroded in simulated seawater strengthened by shape memory alloy wires. 14 specimens were made through the test, and the actual seawater corrosion was simulated by preparing a certain concentration of artificial seawater. FEM was also conducted. This paper can be published after a professional English editor.

Reviewer #2: The authors conducted experimental and numerical studies on the strength degradation properties of SMA-reinforced concrete columns corroded by seawater. But several issues should be addressed before acceptance and publication.

1) The title needs to be more concise such as Study on mechanical properties of SMA-reinforced corroded concrete columns.

2) The literature review of application of SMA wire/cable-based components (e.g., Eng. Struct., 183: 533-549, 2019; Earth. Eng. Struct. Dyn., 2021, https://doi.org/10.1002/eqe.3555.) should be referred.

3) The manuscript needs to be polished by native speakers of English.

4) The numerical simulation results together with the test results are both recommended to provide.

5) The axial force vs. axial deformation of the SMA-reinforced cylinder specimen, shown in Figure 8, should be provided.

Reviewer #3: This paper investigates the behaviours of the concrete columns corroded in seawater. Shape memory alloy wires were applied aiming to increase the strength and energy dissipation capacities. This paper is not well written since the key methodologies were designed groundlessly and were also unclearly presented. The authors are lacking fundamental understandings of the SMA and may use SMA just for creating innovations. Detailed comments are shown in below:

1) Introduction: It is a poor introduction to present why SMA can better improve the strength and the energy dissipation. It should firstly talk more about the material characteristics of the SMA and discuss how these characteristics can help to improve the capacities.

2) Introduction: There are different families in SMAs (Cu-Based, NiTi-based and Fe-based). Different SMAs have different properties and may behave totally different. However, this paper did not mention this.

3) Section 2.1: The naming methods of the specimens were missing.

4) Section 2.1: The grading of concrete is according to some Chinese standards but is lacking citations.

5) Section 2.2: There is no information about the SMA used in this study. Transformation temperatures (usually from DSC), dimensions, phase status, compositions and supplying source were not provided. So, the reliability of the paper is doubted.

6) Section 2.2: How the SMA wires strengthen the concrete column is ambiguous.

7) Section 2.3: Loading protocol is missing.

8) Methods: Poor illustration of the Methods makes me difficultly understand the results.

9) Section 3.1: There were no discussions to explain the failure modes. It looks like an experimental report rather than a scientific paper.

10) Results: It is doubted how the testing programme was designed. It is apparent that reinforced concrete columns should have better capacity than the unreinforced ones no matter which material is used for reinforcement. It should be a comparison between SMA-reinforced concrete column and traditional material-reinforced concrete column. Thus, I cannot find the significance in the results part.

11) Section 4: How was the SMA modelled? These was no material testing on SMA in this paper. How does these properties of the SMA come from? It makes the simulation results not solid.

12) Conclusions: No solid conclusions were drawn, and the experimental problem was not well-formulated.

6. PLOS authors have the option to publish the peer review history of their article (what does this mean?). If published, this will include your full peer review and any attached files.

Reviewer #1: No

Reviewer #2: No

Reviewer #3: No

---

## [Author Response · Author response to Decision Letter 0]

11 Aug 2022

一、Response to the question of reviewer #1

1. This paper inverstigates mechanical properties of reinforced concrete columns corroded in simulated seawater strengthened by shape memory alloy wires. 14 specimens were made through the test, and the actual seawater corrosion was simulated by preparing a certain concentration of artificial seawater. FEM was also conducted. This paper can be published after a professional English editor.

Response : We have asked English professionals to revise the whole manuscript.

二、Response to the question of reviewer #2

1. The title needs to be more concise such as Study on mechanical properties of SMA-reinforced corroded concrete columns.

Response : Thank you for your remind. The title has been changed from "Study on mechanical properties of reinforced concrete columns corroded in simulated seawater strengthened by shape memory alloy wires" to "Study on mechanical properties of corroded concrete columns strengthened with SMA wires".

2. The literature review of application of SMA wire/cable-based components (e.g., Eng. Struct., 183: 533-549, 2019; Earth. Eng. Struct. Dyn., 2021, https://doi.org/10.1002/eqe.3555.) should be referred.

Response : Thank you for your remind. The application literature review of SMA wire has been added in the introduction.

3. The manuscript needs to be polished by native speakers of English.

Response : We have polished the language of the whole manuscript.

4. The numerical simulation results together with the test results are both recommended to provide.

Response :Thank you for your remind. Numerical simulation results and test results have been supplemented.

5. The axial force vs. axial deformation of the SMA-reinforced cylinder specimen, shown in Figure 8, should be provided.

Response : This is a very detailed issue and thank you for your remind. The deformation diagram of SMA wire reinforced specimen has been added at the corresponding position.

三、Response to the question of reviewer #3

1. Introduction: It is a poor introduction to present why SMA can better improve the strength and the energy dissipation. It should firstly talk more about the material characteristics of the SMA and discuss how these characteristics can help to improve the capacities.

Response : Thank you for your detailed reading. It has been revised according to these suggestions in the introduction. 

2. Introduction: There are different families in SMAs (Cu-Based, NiTi-based and Fe-based). Different SMAs have different properties and may behave totally different. However, this paper did not mention this.

Response : Thank you for your remind. In this paper, NiTi-SMA wire is used for reinforcement, which has been explained in the abstract and introduction. Because this paper mainly studies the reinforcement effect of SMA wire on corroded columns, and Ni-Ti SMA wire has the characteristics of hyperelasticity and good deformation ability, which can help us achieve the research purpose.

3. Section 2.1: The naming methods of the specimens were missing.

Response : Thank you for your remind. The specimen naming method has been described. For example, C1W-1 represents the first unreinforced concrete column in the first group.

4. Section 2.1: The grading of concrete is according to some Chinese standards but is lacking citations.

Response : Thank you for your detailed reading. Relevant standards have been quoted.

5. Section 2.2: There is no information about the SMA used in this study. Transformation temperatures (usually from DSC), dimensions, phase status, compositions and supplying source were not provided. So, the reliability of the paper is doubted.

Response : This is a very detailed issue and thank you for your remind. The basic information of SMA used in this test has been completed.

6. Section 2.2: How the SMA wires strengthen the concrete column is ambiguous.

Response : Thank you for your remind. The method of strengthening reinforced concrete columns with SMA wires has been described and supplemented.

7. Section 2.3: Loading protocol is missing.

Response : We have added.

8. Methods: Poor illustration of the Methods makes me difficultly understand the results.

Response : We have revised the loading method and added the loading protocol to help understand the loading method.

9. Section 3.1: There were no discussions to explain the failure modes. It looks like an experimental report rather than a scientific paper.

Response : At the beginning, what is discussed in this section is the phenomenon of the test and the final failure mode of the test piece. In this paper, there is a clear test process and test results, and the whole test is analyzed and the corresponding conclusions are drawn. And after revises, some deficiencies have also been improved. 

10. Results: It is doubted how the testing programme was designed. It is apparent that reinforced concrete columns should have better capacity than the unreinforced ones no matter which material is used for reinforcement. It should be a comparison between SMA-reinforced concrete column and traditional material-reinforced concrete column. Thus, I cannot find the significance in the results part. 

Response : First, the research idea proposed by reviewer is a good idea, which is also the consideration category of our original experimental design. The purpose of this paper is to study the strengthening effect of SMA wire on corroded concrete columns. The comparative analysis of materials can be studied in the next step.

11. Section 4: How was the SMA modelled? These was no material testing on SMA in this paper. How does these properties of the SMA come from? It makes the simulation results not solid.

Response : The SMA modeling process is introduced, and the SMA wire material property test has been supplemented. Thank you very much for your remind.

12. Conclusions: No solid conclusions were drawn, and the experimental problem was not well-formulated.

Response : We revised some conclusions and reached reliable conclusions from qualitative and quantitative aspects respectively. At the same time, the experimental part of the paper is also supplemented to make its conclusion more reasonable.

---

## [Decision Letter · Decision Letter 1]

20 Sep 2022

PONE-D-22-13623R1Study on mechanical properties of corroded concrete columns strengthened with SMA wiresPLOS ONE

Dear Dr. Pei,

Thank you for submitting your manuscript to PLOS ONE. After careful consideration, we feel that it has merit but does not fully meet PLOS ONE’s publication criteria as it currently stands. Therefore, we invite you to submit a revised version of the manuscript that addresses the points raised during the review process.

We look forward to receiving your revised manuscript.

Kind regards,

Cheng Fang

Academic Editor

PLOS ONE

Journal Requirements:

Reviewers' comments:

Reviewer's Responses to Questions

**Comments to the Author**

1. If the authors have adequately addressed your comments raised in a previous round of review and you feel that this manuscript is now acceptable for publication, you may indicate that here to bypass the “Comments to the Author” section, enter your conflict of interest statement in the “Confidential to Editor” section, and submit your "Accept" recommendation.

Reviewer #2: All comments have been addressed

Reviewer #3: All comments have been addressed

2. Is the manuscript technically sound, and do the data support the conclusions?

Reviewer #2: Yes

Reviewer #3: Yes

3. Has the statistical analysis been performed appropriately and rigorously? 

Reviewer #2: Yes

Reviewer #3: Yes

4. Have the authors made all data underlying the findings in their manuscript fully available?

Reviewer #2: (No Response)

Reviewer #3: Yes

5. Is the manuscript presented in an intelligible fashion and written in standard English?

Reviewer #2: Yes

Reviewer #3: Yes

6. Review Comments to the Author

Reviewer #2: There are many typos in the current manuscript such as the unit kn in page 17 should be corrected as kN.

The last sentence of the revise manuscript is wrong and it should be rewritten.

The whole revised manuscript should be checked again carefully.

Reviewer #3: Authors have well revised the paper. Following comments should be considered before publication:

1) More discussions about the future research direction should be involved in the Conclusion or before the Conclusion.

2) A short introduction of different families of SMAs is needed in the Introduction. Following paper should be referred: Chang W-S & Araki Y. Use of shape-memory alloys in construction: a critical review. Proceedings of the Institution of Civil Engineers - Civil Engineering, 2016, 169(2), 87-95.

3) Innovative application of SMA to devices in modern structures is suggested to be reviewed: Huang H, Chang W-S. Re-tuning an off-tuned tuned mass damper by adjusting temperature of shape memory alloy: Exposed to wind action. Structures 2020, 25, 180-189.

7. PLOS authors have the option to publish the peer review history of their article (what does this mean?). If published, this will include your full peer review and any attached files.

Reviewer #2: No

Reviewer #3: No

---

## [Author Response · Author response to Decision Letter 1]

25 Sep 2022

Response to Reviewers

一、Response to the question of reviewer #2

1. There are many typos in the current manuscript such as the unit kn in page 17 should be corrected as kN.

Response : Thank you for your careful reading. We have carefully checked the whole manuscript and revised the typos pointed out by the reviewer.

2. The last sentence of the revise manuscript is wrong and it should be rewritten.

Response : Thank you for your careful reading. We have rewritten the last sentence of the manuscript.

3. The whole revised manuscript should be checked again carefully.

Response : Thank you for your remind. We have carefully checked the whole manuscript and revised the typos.

二、Response to the question of reviewer #3

1. More discussions about the future research direction should be involved in the Conclusion or before the Conclusion.

Response : Thank you for your remind. We have discussed the future research direction in the conclusions and discussions.

2. A short introduction of different families of SMAs is needed in the Introduction. Following paper should be referred: Chang W-S & Araki Y. Use of shape-memory alloys in construction: a critical review. Proceedings of the Institution of Civil Engineers - Civil Engineering, 2016, 169(2), 87-95.

Response : Thank you for your remind. We have introduced different families of SMA in the introduction, and cited this literature.

3. Innovative application of SMA to devices in modern structures is suggested to be reviewed: Huang H, Chang W-S. Re-tuning an off-tuned tuned mass damper by adjusting temperature of shape memory alloy: Exposed to wind action. Structures 2020, 25, 180-189.

Response : Thank you for your remind. In the introduction, we have briefly introduced the innovative application of SMA in modern structural devices, and cited this literature.

Note: This change in the references is to add several references and modify the format of the references.

---

## [Decision Letter · Decision Letter 2]

4 Oct 2022

Study on mechanical properties of corroded concrete columns strengthened with SMA wires

PONE-D-22-13623R2

Dear Dr. Pei,

We’re pleased to inform you that your manuscript has been judged scientifically suitable for publication and will be formally accepted for publication once it meets all outstanding technical requirements.

Kind regards,

Cheng Fang

Academic Editor

PLOS ONE

Additional Editor Comments (optional):

Reviewers' comments:

Reviewer's Responses to Questions

**Comments to the Author**

1. If the authors have adequately addressed your comments raised in a previous round of review and you feel that this manuscript is now acceptable for publication, you may indicate that here to bypass the “Comments to the Author” section, enter your conflict of interest statement in the “Confidential to Editor” section, and submit your "Accept" recommendation.

Reviewer #3: All comments have been addressed

2. Is the manuscript technically sound, and do the data support the conclusions?

Reviewer #3: Yes

3. Has the statistical analysis been performed appropriately and rigorously? 

Reviewer #3: Yes

4. Have the authors made all data underlying the findings in their manuscript fully available?

Reviewer #3: No

5. Is the manuscript presented in an intelligible fashion and written in standard English?

Reviewer #3: Yes

6. Review Comments to the Author

Reviewer #3: This manuscript has been well revised and now is well wiritten. I am satisfied with current manuscript.

7. PLOS authors have the option to publish the peer review history of their article (what does this mean?). If published, this will include your full peer review and any attached files.

Reviewer #3: No

---

## [Editor Report · Acceptance letter]

6 Oct 2022

PONE-D-22-13623R2 

Study on mechanical properties of corroded concrete columns strengthened with SMA wires 

Dear Dr. Pei:

I'm pleased to inform you that your manuscript has been deemed suitable for publication in PLOS ONE. Congratulations! Your manuscript is now with our production department. 

Kind regards, 

on behalf of

Dr. Cheng Fang 

Academic Editor

PLOS ONE